# OMNI-DIMENSIONAL DYNAMIC CONVOLUTION

**Chao Li**[1]\***, Aojun Zhou**[2]**, Anbang Yao**[1]†
[1]Intel Labs China, [2]CUHK-SenseTime Joint Lab, The Chinese University of Hong Kong
`chao.li3@intel.com, aojun.zhou@gmail.com, anbang.yao@intel.com`

## ABSTRACT

Learning a single static convolutional kernel [1] in each convolutional layer is the common training paradigm of modern Convolutional Neural Networks (CNNs). Instead, recent research in dynamic convolution shows that learning a linear combination of $n$ convolutional kernels weighted with their input-dependent attentions can significantly improve the accuracy of light-weight CNNs, while maintaining efficient inference. However, we observe that existing works endow convolutional kernels with the dynamic property through one dimension (regarding the convolutional kernel number) of the kernel space, but the other three dimensions (regarding the spatial size, the input channel number and the output channel number for each convolutional kernel) are overlooked. Inspired by this, we present Omni-dimensional Dynamic Convolution (ODConv), a more generalized yet elegant dynamic convolution design, to advance this line of research. ODConv leverages a novel multi-dimensional attention mechanism with a parallel strategy to learn complementary attentions for convolutional kernels along all four dimensions of the kernel space at any convolutional layer. As a drop-in replacement of regular convolutions, ODConv can be plugged into many CNN architectures. Extensive experiments on the ImageNet and MS-COCO datasets show that ODConv brings solid accuracy boosts for various prevailing CNN backbones including both light-weight and large ones, e.g., 3.77%∼5.71%|1.86%∼3.72% absolute top-1 improvements to MobivleNetV2|ResNet family on the ImageNet dataset. Intriguingly, thanks to its improved feature learning ability, ODConv with even one single kernel can compete with or outperform existing dynamic convolution counterparts with multiple kernels, substantially reducing extra parameters. Furthermore, ODConv is also superior to other attention modules for modulating the output features or the convolutional weights. Code and models will be available at `https://github.com/OSVAI/ODConv`.

## 1 INTRODUCTION

In the past decade, we have witnessed the tremendous success of deep Convolutional Neural Networks (CNNs) in many computer vision applications (Krizhevsky et al., 2012; Girshick et al., 2014; Long et al., 2015; He et al., 2017). The most common way of constructing a deep CNN is to stack a number of convolutional layers as well as other basic layers organized with the predefined feature connection topology. Along with great advances in CNN architecture design by manual engineering (Krizhevsky et al., 2012; He et al., 2016; Howard et al., 2017) and automatic searching (Zoph & Le, 2017; Pham et al., 2018; Howard et al., 2019), lots of prevailing classification backbones have been presented. Recent works (Wang et al., 2017; Hu et al., 2018b; Park et al., 2018; Woo et al., 2018; Yang et al., 2019; Chen et al., 2020) show that incorporating attention mechanisms into convolutional blocks can further push the performance boundaries of modern CNNs, and thus it has attracted great research interest in the deep learning community.

The well-known SENet (Hu et al., 2018b) uses an attention module consisting of squeeze and excitation operations to adaptively recalibrate the output features of convolutional layers, strengthening

---

\*This work was done when Chao Li was an intern at Intel Labs China, supervised by Anbang Yao who proposed the original idea and led the writing of the paper. † Corresponding author.

[1]Here, we follow the definitions in (Yang et al., 2019; Chen et al., 2020) where a convolutional kernel refers to the filter set of a convolutional layer.

the representation power of a CNN via encouraging informative feature channels while suppressing less important ones. Numerous attentive feature recalibration variants (Woo et al., 2018; Park et al., 2018; Hu et al., 2018a) have been proposed since then. In Lin et al. (2020) and Quader et al. (2020), two attention extensions to modulate the convolutional weights instead of the output features are also presented. Unlike the aforementioned methods in which the number of convolutional parameters of a target network is fixed, dynamic convolution, which applies the attention mechanism over $n$ additive convolutional kernels to increase the size and the capacity of a network while maintaining efficient inference, has recently become popular in optimizing efficient CNNs. This line of research is pioneered by Conditionally Parameterized Convolutions (CondConv) (Yang et al., 2019) and Dynamic Convolution (DyConv) (Chen et al., 2020) whose basic ideas are the same. Generally, unlike a regular convolutional layer which applies the same (i.e., static) convolutional kernel to all input samples, a dynamic convolutional layer learns a linear combination of $n$ convolutional kernels weighted with their attentions conditioned on the input features. Despite significant accuracy improvements for light-weight CNNs, dynamic convolution designed in this way has two limitations. Firstly, the main limitation lies in the attention mechanism design. The dynamic property of CondConv and DyConv comes from computing convolutional kernels as a function of the input features. However, we observe that they endow the dynamic property to convolutional kernels through one dimension (regarding the convolutional kernel number) of the kernel space while the other three dimensions (regarding the spatial size, the input channel number and the output channel number for each convolutional kernel) are overlooked. As a result, the weights of each convolutional kernel share the same attention scalar for a given input, limiting their abilities to capture rich contextual cues. That is, the potentials of dynamic convolutional property have not been fully explored by existing works, and thus they leave considerable room for improving the model performance. Secondly, at a convolutional layer, replacing regular convolution by dynamic convolution increases the number of convolutional parameters by $n$ times. When applying dynamic convolution to a lot of convolutional layers, it will heavily increase the model size. To handle this limitation, Li et al. (2021) proposes a dynamic convolution decomposition method which can get more compact yet competitive models. *Instead, in this paper we address both of the above limitations in a new perspective: formulating a more diverse and effective attention mechanism and inserting it into the convolutional kernel space.*

Our core contribution is a more generalized yet elegant dynamic convolution design called Omni-dimensional Dynamic Convolution (ODConv). Unlike existing works discussed above, at any convolutional layer, ODConv leverages a novel multi-dimensional attention mechanism to learn four types of attentions for convolutional kernels along all four dimensions of the kernel space in a parallel manner. We show that these four types of attentions learnt by our ODConv are complementary to each other, and progressively applying them to the corresponding convolutional kernels can substantially strengthen the feature extraction ability of basic convolution operations of a CNN. Consequently, ODConv with even one single kernel can compete with or outperform existing dynamic convolution counterparts with multiple kernels, substantially reducing extra parameters.

As a drop-in design, ODConv can be used to replace regular convolutions in many CNN architectures. It strikes a better tradeoff between model accuracy and efficiency compared to existing dynamic convolution designs, as validated by extensive experiments on the large-scale ImageNet classification dataset (Russakovsky et al., 2015) with various prevailing CNN backbones. ODConv also shows better recognition performance under similar model complexities when compared to other state-of-the-art attention methods for output feature recalibration (Woo et al., 2018; Hu et al., 2018b; Wang et al., 2020; Lin et al., 2020) or for convolutional weight modification (Ma et al., 2020; Lin et al., 2020; Quader et al., 2020). Furthermore, the performance improvements by ODConv for the pre-trained classification models can transfer well to downstream tasks such as object detection on the MS-COCO dataset (Lin et al., 2014), validating its promising generalization ability.

## 2 RELATED WORK

**Deep CNN Architectures.** AlexNet (Krizhevsky et al., 2012) ignited the surge of deep CNNs by winning the ImageNet classification challenge 2012. Since then, lots of well-known CNN architectures such as VGGNet (Simonyan & Zisserman, 2015), InceptionNet (Szegedy et al., 2015), ResNet (He et al., 2016), DenseNet (Huang et al., 2017) and ResNeXt (Xie et al., 2017) have been proposed, which are designed to be much deeper and have more sophisticated connection topologies compared with AlexNet. To ease the deployment of inference models on resource-limited platforms,

MobileNets (Howard et al., 2017; Sandler et al., 2018) and ShuffleNet (Zhang et al., 2018b; Ma et al., 2018) are presented. All the aforementioned CNNs are manually designed. Recently, researchers have also made great efforts (Zoph & Le, 2017; Pham et al., 2018; Howard et al., 2019) to automate the network design process. Our ODConv could be potentially used to boost their performance.

**Attentive Feature Recalibration.** Designing attentive feature recalibration modules to improve the performance of a CNN has been widely studied in recent years. Wang et al. (2017) proposes a specialized attention module consisting of a trunk branch and a mask branch, and inserts it into the intermediate stages of deep residual networks. SENet (Hu et al., 2018b) uses a seminal channel attention module termed Squeeze-and-Excitation (SE) to exploit the interdependencies between the channels of convolutional features. Many subsequent works improve SE from different aspects, following its two-stage design (i.e., feature aggregation and feature recalibration). BAM (Park et al., 2018) and CBAM (Woo et al., 2018) combine the channel attention module with the spatial attention module. Misra et al. (2021) presents an attention module having three branches conditioned on the features rotated along three different dimensions. GE (Hu et al., 2018a) introduces a gather operator to extract better global context from a large spatial extent. To enhance the feature aggregation capability, SRM (Lee et al., 2019) replaces the global average by the channel-wise mean and standard deviation. SKNets (Li et al., 2019) add an attention design over two branches with different sized convolutions to fuse multi-scale feature outputs. ECA (Wang et al., 2020) provides a more efficient channel attention design using cheaper 1D convolutions to replace the first fully connected layer of SE. Instead of recalibrating the output convolutional features by attention modules, dynamic convolution methods apply attention mechanisms to a linear combination of $n$ convolutional kernels.

**Dynamic Weight Networks.** Making the weights of a neural network to be sample-adaptive via dynamic mechanisms has shown great potentials for boosting model capacity and generalization. Hypernetworks (Ha et al., 2017) use a small network called hypernetwork to generate the weights for a larger recurrent network called main network. MetaNet (Munkhdalai & Yu, 2017) adopts a meta learning model to parameterize the task-adaptive network for rapid generalization across a sequence of tasks. Jaderberg et al. (2015) proposes a Spatial Transformer module conditioned on the learnt features to predict the parametric transformation, and applies it to align the distorted input image. Dynamic Filter Network (Jia et al., 2016) uses a filter generation network to produce filters conditioned on an input, and processes another input with the generated filters. DynamoNet (Diba et al., 2019) uses dynamically generated motion filters to handle the action recognition problem. Kernel Prediction Networks (Bako et al., 2017; Mildenhall et al., 2018) leverage a CNN architecture to predict spatially varying kernels used for video denoising. WeightNet (Ma et al., 2020) appends a grouped fully connected layer to the attention feature vector of an SE block, generating the weights of a CNN used for image recognition. Lin et al. (2020) modifies the weights of convolutional layers with a gated module under the guidance of global context, while Quader et al. (2020) directly uses either an SE block or a simple activation function conditioned on the magnitudes of convolutional weights to modify the weights themselves. Our ODConv aims to address the limitations of recently proposed dynamic convolution (Yang et al., 2019; Chen et al., 2020) which differs from these methods both in focus and formulation, see the Introduction and Method sections for details.

## 3 METHOD

In this section, we first make a review of dynamic convolution via a general formulation. Then, we describe the formulation of our ODConv, clarify its properties and detail its implementation.

### 3.1 REVIEW OF DYNAMIC CONVOLUTION

**Basic concept.** A regular convolutional layer has a single static convolutional kernel which is applied to all input samples. For a dynamic convolutional layer, it uses a linear combination of $n$ convolutional kernels weighted dynamically with an attention mechanism, making convolution operations be input-dependent. Mathematically, dynamic convolution operations can be defined as

$$y = (\alpha_{w1}W_1 + ... + \alpha_{wn}W_n) * x, \tag{1}$$

where $x \in \mathbb{R}^{h \times w \times c_{in}}$ and $y \in \mathbb{R}^{h \times w \times c_{out}}$ denote the input features and the output features (having $c_{in}/c_{out}$ channels with the height $h$ and the width $w$), respectively; $W_i$ denotes the $i^{th}$ convolutional kernel consisting of $c_{out}$ filters $W_i^m \in \mathbb{R}^{k \times k \times c_{in}}, m = 1, ..., c_{out}$; $\alpha_{wi} \in \mathbb{R}$ is the attention scalar

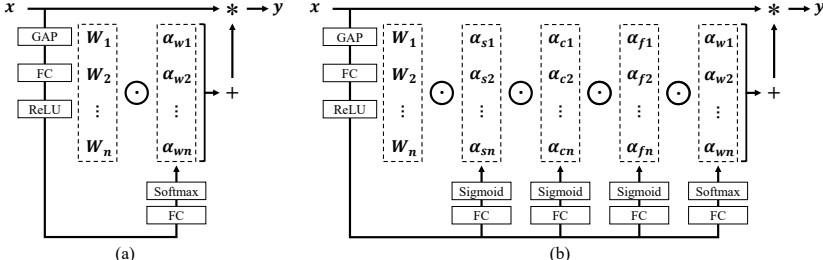

Figure 1: A schematic comparison of (a) DyConv (CondConv uses GAP+FC+Sigmoid) and (b) ODConv. Unlike CondConv and DyConv which compute a single attention scalar $\alpha_{wi}$ for the convolutional kernel $W_i$, ODConv leverages a novel multi-dimensional attention mechanism to compute four types of attentions $\alpha_{si}$, $\alpha_{ci}$, $\alpha_{fi}$ and $\alpha_{wi}$ for $W_i$ along all four dimensions of the kernel space in a parallel manner. Their formulations and implementations are clarified in the Method section.

for weighting $W_i$, which is computed by an attention function $\pi_{wi}(x)$ conditioned on the input features; $*$ denotes the convolution operation. For conciseness, here we omit the bias term.

**CondConv vs. DyConv.** Although the concept of dynamic convolution defined in Eq. 1 is proposed separately in CondConv (Yang et al., 2019) and DyConv (Chen et al., 2020), their implementations are different, mainly in the structure of $\pi_{wi}(x)$ to compute $\alpha_{wi}$, the model training strategy, and the layer locations to apply dynamic convolutions. Specifically, both methods choose the modified SE structure for $\pi_{wi}(x)$, and CondConv uses a Sigmoid function while DyConv uses a Softmax function as the activation function to compute $\alpha_{wi}$. DyConv adopts a temperature annealing strategy in the training process to suppress the near one-hot output of the Softmax function. For all their tested CNN architectures, CondConv replaces the convolutional layers in the final several blocks (e.g., 6 for the MobileNetV2 backbones and 3 for the ResNet backbones) and the last fully connected layer, while DyConv replaces all convolutional layers except the first layer. These implementation differences lead to different results in model accuracy, size and efficiency for CondConv and DyConv.

**Limitation Discussions.** According to Eq. 1, dynamic convolution has two basic components: the convolutional kernels $\{W_1, ...W_n\}$, and the attention function $\pi_{wi}(x)$ to compute their attention scalars $\{\alpha_{w1}, ...\alpha_{wn}\}$. Given $n$ convolutional kernels, the corresponding kernel space has four dimensions regarding the spatial kernel size $k \times k$, the input channel number $c_{in}$ and the output channel number $c_{out}$ for each convolutional kernel, and the convolutional kernel number $n$. However, for CondConv and DyConv, we can observe that $\pi_{wi}(x)$ allocates a single attention scalar $\alpha_{wi}$ to the convolutional kernel $W_i$, meaning that all its $c_{out}$ filters $W_i^m \in \mathbb{R}^{k \times k \times c_{in}}$, $m = 1, ..., c_{out}$ have the same attention value for the input $x$. In other words, the spatial dimension, the input channel dimension and the output channel dimension for the convolutional kernel $W_i$ are ignored by CondConv and DyConv. This leads to a coarse exploitation of the kernel space when they design their attention mechanisms for endowing $n$ convolutional kernels with the dynamic property. This may also be one of the reasons why CondConv and DyConv show much lower performance gains to relatively larger CNNs compared to efficient ones. Besides, compared to a regular convolutional layer, a dynamic convolutional layer increases the number of convolutional parameters by $n$ times (although the increase of Multiply-Adds (MAdds) is marginal due to the additive property of $n$ convolutional kernels). Typically, CondConv sets $n = 8$ and DyConv sets $n = 4$. Therefore, it will heavily increase the model size when applying dynamic convolution to a lot of convolutional layers. However, we empirically find that removing the attention mechanism from CondConv|DyConv (i.e., setting $\alpha_{wi} = 1$) almost diminishes the accuracy boosts for prevailing CNN backbones on the ImageNet dataset close to zero. For instance, on ResNet18, the top-1 gain averaged over 3 runs decreases from $1.74\%|2.51\%$ to $0.08\%|0.14\%$ when removing the attention mechanism from CondConv|DyConv. *These observations indicate that the attention mechanism design plays the key role in dynamic convolution, and a more effective design may strike a good balance between model accuracy and size.*

### 3.2 Omni-Dimensional Dynamic Convolution

In light of the above discussions, our ODConv introduces a multi-dimensional attention mechanism with a parallel strategy to learn diverse attentions for convolutional kernels along all four dimensions of the kernel space. Fig. 1 provides a schematic comparison of CondConv, DyConv and ODConv.

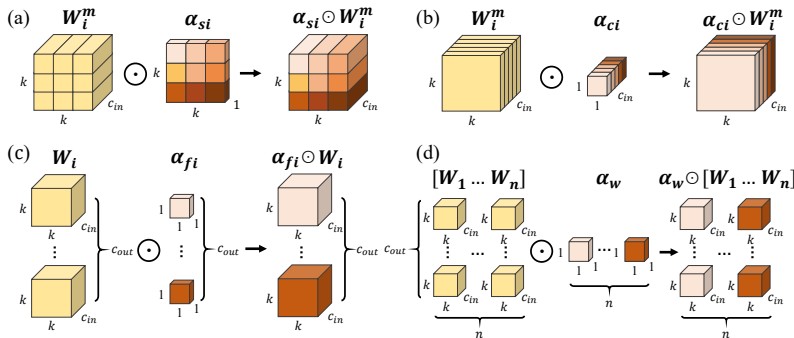

Figure 2: Illustration of multiplying four types of attentions in ODConv to convolutional kernels progressively. (a) Location-wise multiplication operations along the spatial dimension, (b) channel-wise multiplication operations along the input channel dimension, (c) filter-wise multiplication operations along the output channel dimension, and (d) kernel-wise multiplication operations along the kernel dimension of the convolutional kernel space. Notations are clarified in the Method section.

**Formulation of ODConv.** Following the notations in Eq. 1, ODConv can be defined as

$$y = (\alpha_{w1} \odot \alpha_{f1} \odot \alpha_{c1} \odot \alpha_{s1} \odot W_1 + ... + \alpha_{wn} \odot \alpha_{fn} \odot \alpha_{cn} \odot \alpha_{sn} \odot W_n) * x, \quad (2)$$

where $\alpha_{wi} \in \mathbb{R}$ denotes the attention scalar for the convolutional kernel $W_i$, which is the same to that in Eq. 1; $\alpha_{si} \in \mathbb{R}^{k \times k}$, $\alpha_{ci} \in \mathbb{R}^{c_{in}}$ and $\alpha_{fi} \in \mathbb{R}^{c_{out}}$ denote three newly introduced attentions, which are computed along the spatial dimension, the input channel dimension and the output channel dimension of the kernel space for the convolutional kernel $W_i$, respectively; $\odot$ denotes the multiplication operations along different dimensions of the kernel space. Here, $\alpha_{si}$, $\alpha_{ci}$, $\alpha_{fi}$ and $\alpha_{wi}$ are computed with a multi-head attention module $\pi_i(x)$ which will be clarified later.

**A Deep Understanding of ODConv.** In ODConv, for the convolutional kernel $W_i$: (1) $\alpha_{si}$ assigns different attention scalars to convolutional parameters (per filter) at $k \times k$ spatial locations; (2) $\alpha_{ci}$ assigns different attention scalars to $c_{in}$ channels of each convolutional filter $W_i^m$; (3) $\alpha_{fi}$ assigns different attention scalars to $c_{out}$ convolutional filters; (4) $\alpha_{wi}$ assigns an attention scalar to the whole convolutional kernel. Fig. 2 illustrates the process of multiplying these four types of attentions to $n$ convolutional kernels. *In principle, these four types of attentions are complementary to each other, and progressively multiplying them to the convolutional kernel $W_i$ in the location-wise, channel-wise, filter-wise and kernel-wise orders makes convolution operations be different w.r.t. all spatial locations, all input channels, all filters and all kernels for the input $x$, providing a performance guarantee to capture rich context cues.* Therefore, ODConv can significantly strengthen the feature extraction ability of basic convolution operations of a CNN. Moreover, ODConv with one single convolutional kernel can compete with or outperform standard CondConv and DyConv, introducing substantially fewer extra parameters to the final models. Extensive experiments are provided to validate these advantages. By comparing Eq. 1 and Eq. 2, we can clearly see that ODConv is a more generalized dynamic convolution. Moreover, when setting $n = 1$ and all components of $\alpha_{s1}$, $\alpha_{c1}$ and $\alpha_{w1}$ to 1, ODConv with only filter-wise attention $\alpha_{f1}$ will be reduced into: applying an SE variant conditioned on the input features to the convolutional filters, then followed by convolution operations (note the original SE (Hu et al., 2018b) is conditioned on the output features, and is used to recalibrate the output features themselves). Such an SE variant is a special case of ODConv.

**Implementation.** For ODConv, a critical question is how to compute four types of attentions $\alpha_{si}$, $\alpha_{ci}$, $\alpha_{fi}$ and $\alpha_{wi}$ for the convolutional kernel $W_i$. Following CondConv and DyConv, we also use a SE-typed attention module (Hu et al., 2018b) but with multiple heads as $\pi_i(x)$ to compute them, whose structure is shown in Fig. 1. Specifically, the input $x$ is squeezed into a feature vector with the length of $c_{in}$ by channel-wise Global Average Pooling (GAP) operations first. Subsequently, there is a Fully Connected (FC) layer and four head branches. A Rectified Linear Unit (ReLU) (Krizhevsky et al., 2012) comes after the FC layer. The FC layer maps the squeezed feature vector to a lower dimensional space with the reduction ratio $r$ (according to the ablative experiments, we set $r = 1/16$ in all main experiments, avoiding high model complexity). For four head branches, each has an FC layer with the output size of $k \times k$, $c_{in} \times 1$, $c_{out} \times 1$ and $n \times 1$, and a Softmax or Sigmoid function to generate the normalized attentions $\alpha_{si}$, $\alpha_{ci}$, $\alpha_{fi}$ and $\alpha_{wi}$, respectively. We adopt the temperature annealing strategy proposed in DyConv to facilitate the training process. *For easy implementation, we apply ODConv to all convolutional layers except the first layer of each CNN architecture tested*

*in our main experiments just like DyConv, and share $\alpha_{si}$, $\alpha_{ci}$ and $\alpha_{fi}$ to all convolutional kernels.* In the Experiments section, a comparison of the inference speed for different dynamic convolution methods is provided. In the Appendix, we further provide a computational cost analysis of ODConv and an ablation study of applying ODConv to different layer locations.

## 4 EXPERIMENTS

In this section, we provide comprehensive experiments on two large-scale image recognition datasets with different CNN architectures to validate the effectiveness of ODConv, compare its performance with many attention based methods, and study the design of ODConv from different aspects.

### 4.1 IMAGE CLASSIFICATION ON IMAGENET

Our main experiments are performed on the ImageNet dataset (Russakovsky et al., 2015). It has over 1.2 million images for training and 50,000 images for validation, including 1,000 object classes.

**CNN Backbones.** We use MobileNetV2 (Sandler et al., 2018) and ResNet (He et al., 2016) families for experiments, covering both light-weight CNN architectures and larger ones. Specifically, we choose ResNet18, ResNet50, ResNet101, and MobileNetV2 ($1.0\times$, $0.75\times$, $0.5\times$) as the backbones.

**Experimental Setup.** In the experiments, we consider existing dynamic convolution methods including CondConv (Yang et al., 2019), DyConv (Chen et al., 2020) and DCD (Li et al., 2021) as the key reference methods for comparisons on all CNN backbones. On the ResNet backbones, we also compare our ODConv with many state-of-the-art methods using attention modules: (1) for output feature recalibration including SE (Hu et al., 2018b), CBAM (Woo et al., 2018) and ECA (Wang et al., 2020), and (2) for convolutional weight modification including CGC (Lin et al., 2020), Weight-Net (Ma et al., 2020) and WE (Quader et al., 2020). *For fair comparisons, we use public codes of these methods unless otherwise stated, and adopt the popular training and test settings used in the community for implementing the experiments. The models trained by all methods use the same settings including the batch size, the number of training epochs, the learning rate schedule, the weight decay, the momentum and the data processing pipeline. Moreover, we do not use advanced training tricks such as mixup (Zhang et al., 2018a) and label smoothing (Szegedy et al., 2016), aiming to have clean performance comparisons.* Experimental details are described in the Appendix.

Table 1: Results comparison on the ImageNet validation set with the MobileNetV2 ($1.0\times$, $0.75\times$, $0.5\times$) backbones trained for 150 epochs. For our ODConv, we set $r = 1/16$. Best results are bolded.

| Models | Params | MAdds | Top-1 Acc (%) | Top-5 Acc (%) |
|---|---|---|---|---|
| MobileNetV2 ($1.0\times$) | 3.50M | 300.8M | 71.65 | 90.22 |
| + CondConv ($8\times$) | 22.88M | 318.1M | 74.13 (↑2.48) | 91.67 (↑1.45) |
| + DyConv ($4\times$) | 12.40M | 317.1M | 74.94 (↑3.29) | 91.83 (↑1.61) |
| + DCD | 5.72M | 318.4M | 74.18 (↑2.53) | 91.72 (↑1.50) |
| + ODConv ($1\times$) | 4.94M | 311.8M | 74.84 (↑3.19) | 92.13 (↑1.91) |
| + ODConv ($4\times$) | 11.52M | 327.1M | **75.42 (↑3.77)** | **92.18 (↑1.96)** |
| MobileNetV2 ($0.75\times$) | 2.64M | 209.1M | 69.18 | 88.82 |
| + CondConv ($8\times$) | 17.51M | 223.9M | 71.79 (↑2.61) | 90.17 (↑1.35) |
| + DyConv ($4\times$) | 7.95M | 220.1M | 72.75 (↑3.57) | 90.93 (↑2.11) |
| + DCD | 4.08M | 222.9M | 71.92 (↑2.74) | 90.20 (↑1.38) |
| + ODConv ($1\times$) | 3.51M | 217.1M | 72.43 (↑3.25) | 90.82 (↑2.00) |
| + ODConv ($4\times$) | 7.50M | 226.3M | **73.81 (↑4.63)** | **91.33 (↑2.51)** |
| MobileNetV2 ($0.5\times$) | 2.00M | 97.1M | 64.30 | 85.21 |
| + CondConv ($8\times$) | 13.61M | 110.0M | 67.24 (↑2.94) | 87.51 (↑2.30) |
| + DyConv ($4\times$) | 4.57M | 103.2M | 69.05 (↑4.75) | 88.37 (↑3.16) |
| + DCD | 3.06M | 105.6M | 69.32 (↑5.02) | 88.44 (↑3.23) |
| + ODConv ($1\times$) | 2.43M | 101.8M | 68.26 (↑3.96) | 87.98 (↑2.77) |
| + ODConv ($4\times$) | 4.44M | 106.4M | **70.01 (↑5.71)** | **89.01 (↑3.80)** |

**Results Comparison on MobileNets.** Table 1 shows the results comparison on the MobileNetV2 ($1.0\times$, $0.75\times$, $0.5\times$) backbones. As CondConv and DyConv are primarily proposed to improve the performance of efficient CNNs, they all bring promising top-1 gains to the light-weight MobileNetV2 ($1.0\times$, $0.75\times$, $0.5\times$) backbones. Comparatively, our ODConv ($1\times$) with one single convolutional kernel performs better than CondConv ($8\times$) with 8 convolutional kernels, and its performance is also on par with DyConv ($4\times$) with 4 convolutional kernels. Note that these competitive results of ODConv ($1\times$) are obtained with significantly lower numbers of extra parameters, validating that our ODConv can strike a better tradeoff between model accuracy and size. Besides, ODConv ($1\times$) also performs better than DCD in most cases. ODConv ($4\times$) always achieves the best results

on all backbones. The results comparison with an increase number of training epochs (from 150 to 300) can be found in the Appendix, from which we can observe similar performance trends.

**Results Comparison on ResNets.** Table 2 shows the results comparison on the ResNet18 and ResNet50 backbones which are much larger than the MobileNetV2 backbones. We can obtain the following observations: (1) On the ResNet18 backbone, dynamic convolution methods (CondConv, DyConv, DCD and our ODConv) and convolutional weight modification methods (CGC, WeightNet and WE) mostly show better performance than output feature recalibration methods (SE, CBAM and ECA), although they all use attention mechanisms. Comparatively, our ODConv $(1\times)$ with one single convolutional kernel outperforms the other methods both in model accuracy and size, bringing 2.85% top-1 gain to the baseline model. ODConv $(4\times)$ gets the best results, yielding a top-1 gain of 3.72%; (2) However, on the larger ResNet50 backbone, CondConv, DyConv and DCD show worse results than most of the other methods even though they have significantly increased numbers of parameters. Due to the increased parameter redundancy, adding more parameters to larger networks tends to be less effective in improving model accuracy, compared to small networks. Thanks to the proposed multi-dimensional attention mechanism, ODConv can address this problem better, achieving superior performance both in model accuracy and size for larger backbones as well as light-weight ones. To further validate the performance of ODConv on very deep and large networks, we apply ODConv to the ResNet101 backbone, and the results comparison is provided in Table 3. Again, ODConv shows very promising results, yielding 1.57% top-1 gain with ODConv $(1\times)$.

Table 2: Results comparison on the ImageNet validation set with the ResNet18 and ResNet50 backbones trained for 100 epochs. For our ODConv, we set $r = 1/16$. * denotes the results are from the paper of WE (Quader et al., 2020) as its code is not publicly available. Best results are bolded.

| Network | ResNet18 | | | | ResNet50 | | | |
|---|---|---|---|---|---|---|---|---|
| Models | Params | MAdds | Top-1 Acc (%) | Top-5 Acc (%) | Params | MAdds | Top-1 Acc (%) | Top-5 Acc (%) |
| Baseline | 11.69M | 1.814G | 70.25 | 89.38 | 25.56M | 3.858G | 76.23 | 93.01 |
| + CondConv $(8\times)$ | 81.35M | 1.894G | 71.99 (↑1.74) | 90.27 (↑0.89) | 129.86M | 3.978G | 76.70 (↑0.47) | 93.12 (↑0.11) |
| + DyConv $(4\times)$ | 45.47M | 1.861G | 72.76 (↑2.51) | 90.79 (↑1.41) | 100.88M | 3.965G | 76.82 (↑0.59) | 93.16 (↑0.15) |
| + DCD | 14.70M | 1.841G | 72.33 (↑2.08) | 90.65 (↑1.27) | 29.84M | 3.944G | 76.92 (↑0.69) | 93.46 (↑0.45) |
| + ODConv $(1\times)$ | 11.94M | 1.838G | 73.10 (↑2.85) | 91.10 (↑1.72) | 28.64M | 3.916G | 77.96 (↑1.73) | 93.84 (↑0.83) |
| + ODConv $(4\times)$ | 44.90M | 1.916G | **73.97 (↑3.72)** | **91.35 (↑1.97)** | 90.67M | 4.080G | **78.52 (↑2.29)** | **94.01 (↑1.00)** |
| + SE | 11.78M | 1.816G | 70.98 (↑0.73) | 90.03 (↑0.65) | 28.07M | 3.872G | 77.31 (↑1.08) | 93.63 (↑0.62) |
| + CBAM | 11.78M | 1.818G | 71.01 (↑0.76) | 89.85 (↑0.47) | 28.07M | 3.886G | 77.46 (↑1.23) | 93.59 (↑0.58) |
| + ECA | 11.69M | 1.816G | 70.60 (↑0.35) | 89.68 (↑0.30) | 25.56M | 3.870G | 77.34 (↑1.11) | 93.64 (↑0.63) |
| + CGC | 11.69M | 1.827G | 71.60 (↑1.35) | 90.35 (↑0.97) | 25.59M | 3.877G | 76.79 (↑0.56) | 93.37 (↑0.36) |
| + WeightNet | 11.93M | 1.826G | 71.56 (↑1.31) | 90.38 (↑1.00) | 30.38M | 3.885G | 77.51 (↑1.28) | 93.69 (↑0.68) |
| + WE (*) | 11.90M | 1.820G | 71.00 (↑0.75) | 90.00 (↑0.62) | 28.10M | 3.860G | 77.10 (↑0.87) | 93.50 (↑0.49) |

Table 3: Results comparison on the ImageNet validation set with the ResNet101 backbone trained for 100 epochs. For our ODConv, we set $r = 1/16$. Best results are bolded.

| Models | Params | MAdds | Top-1 Acc (%) | Top-5 Acc (%) |
|---|---|---|---|---|
| ResNet101 | 44.55M | 7.570G | 77.41 | 93.67 |
| + SE | 49.29M | 7.593G | 78.42 (↑1.01) | 94.15 (↑0.48) |
| + CBAM | 49.30M | 7.617G | 78.50 (↑1.09) | 94.20 (↑0.53) |
| + ECA | 44.55M | 7.590G | 78.60 (↑1.19) | 94.34 (↑0.67) |
| + ODConv $(1\times)$ | 50.82M | 7.675G | 78.98 (↑1.57) | 94.38 (↑0.71) |
| + ODConv $(2\times)$ | 90.44M | 7.802G | **79.27 (↑1.86)** | **94.47 (↑0.80)** |

## 4.2 Object Detection on MS-COCO

Given a backbone model pre-trained on the ImageNet classification dataset, a critical question is whether the performance improvement by ODConv can be transferred to downstream tasks or not. To explore it, we next perform comparative experiments on the object detection track of MS-COCO dataset (Lin et al., 2014). The 2017 version of MS-COCO dataset contains 118,000 training images and 5,000 validation images with 80 object classes.

**Experimental Setup.** We use the popular MMDetection toolbox (Chen et al., 2019) for experiments with the pre-trained ResNet50 and MobileNetV2 $(1.0\times)$ models as the backbones for the detector. We select the mainstream Faster R-CNN (Ren et al., 2015) and Mask R-CNN (He et al., 2017) detectors with Feature Pyramid Networks (FPNs) (Lin et al., 2017) as the necks to build the basic object detection systems. For a neat comparison, the convolutional layers in the FPN necks still use regular convolutions, and we maintain the same data preparation pipeline and hyperparameter

Table 4: Results comparison on the MS-COCO 2017 validation set. Regarding Params or MAdds, the number in the bracket is for the pre-trained backbone models excluding the last fully connected layer, which is almost the same to that shown in Table 1 and Table 2, while the other number is for the whole object detector. Best results are bolded.

| Backbone Models | Detectors | $AP(\%)$ | $AP_{50}(\%)$ | $AP_{75}(\%)$ | $AP_S(\%)$ | $AP_M(\%)$ | $AP_L(\%)$ | Params | MAdds |
|---|---|---|---|---|---|---|---|---|---|
| ResNet50 | | 37.2 | 57.8 | 40.4 | 21.5 | 40.6 | 48.0 | 43.80M (23.51M) | 207.07G (76.50G) |
| + CondConv (8×) | | 38.1 | 58.9 | 41.5 | 22.4 | 42.1 | 48.7 | 133.75M (113.46M) | 207.08G (76.51G) |
| + DyConv (4×) | | 38.3 | 59.7 | 41.6 | 22.3 | 42.3 | 49.4 | 119.12M (98.83M) | 207.23G (76.66G) |
| + DCD | | 38.1 | 59.3 | 41.3 | 21.9 | 42.0 | 49.5 | 48.08M (27.79M) | 207.20G (76.63G) |
| + ODConv (1×) | | 39.0 | 60.5 | 42.3 | 23.4 | 42.3 | 50.5 | 46.88M (26.59M) | 207.18G (76.61G) |
| + ODConv (4×) | Faster R-CNN | 39.2 | 60.7 | 42.6 | 23.1 | 42.6 | 51.0 | 108.91M (88.62M) | 207.42G (76.85G) |
| MobileNetV2 (1.0×) | | 31.3 | 51.1 | 33.1 | 17.4 | 33.5 | 41.2 | 21.13M (2.22M) | 122.58G (24.45G) |
| + CondConv (8×) | | 33.7 | 54.9 | 35.6 | 19.3 | 36.4 | 43.7 | 31.54M (12.63M) | 122.59G (24.46G) |
| + DyConv (4×) | | 34.5 | 55.6 | 36.5 | 19.8 | 37.3 | 44.7 | 30.02M (11.12M) | 123.01G (24.88G) |
| + DCD | | 33.3 | 53.0 | 35.1 | 19.9 | 36.1 | 43.2 | 23.34M (4.44M) | 123.01G (24.88G) |
| + ODConv (1×) | | 34.3 | 55.6 | 36.5 | 20.7 | 37.3 | 44.5 | 22.56M (3.66M) | 123.00G (24.87G) |
| + ODConv (4×) | | 35.1 | 56.7 | 37.0 | 20.6 | 38.0 | 45.2 | 29.14M (10.24M) | 123.02G (24.89G) |
| ResNet50 | | 38.0 | 58.6 | 41.5 | 21.6 | 41.5 | 49.2 | 46.45M (23.51M) | 260.14G (76.50G) |
| + CondConv (8×) | | 38.8 | 59.3 | 42.3 | 22.5 | 42.5 | 50.3 | 136.4M (113.46M) | 260.15G (76.51G) |
| + DyConv (4×) | | 39.2 | 60.3 | 42.5 | 23.0 | 42.9 | 51.4 | 121.77M (98.83M) | 260.30G (76.66G) |
| + DCD | | 38.8 | 59.8 | 42.2 | 22.7 | 42.7 | 49.8 | 50.73M (27.79M) | 260.27G (76.63G) |
| + ODConv (1×) | | 39.9 | 61.2 | 43.5 | 23.6 | 43.8 | 52.3 | 49.53M (26.59M) | 260.25G (76.61G) |
| + ODConv (4×) | Mask R-CNN | 40.1 | 61.5 | 43.6 | 24.0 | 43.6 | 52.3 | 111.56M (88.62M) | 260.49G (76.85G) |
| MobileNetV2 (1.0×) | | 32.2 | 52.1 | 34.2 | 18.4 | 34.4 | 42.4 | 23.78M (2.22M) | 175.66G (24.45G) |
| + CondConv (8×) | | 34.4 | 55.4 | 36.6 | 19.8 | 36.9 | 44.6 | 34.19M (12.63M) | 175.67G (24.46G) |
| + DyConv (4×) | | 35.2 | 56.2 | 37.5 | 20.7 | 38.0 | 45.5 | 32.68M (11.12M) | 176.09G (24.88G) |
| + DCD | | 34.3 | 54.9 | 36.6 | 20.6 | 37.1 | 44.8 | 26.00M (4.44M) | 176.09G (24.88G) |
| + ODConv (1×) | | 35.0 | 56.1 | 37.3 | 19.9 | 37.7 | 46.2 | 25.22M (3.66M) | 176.08G (24.87G) |
| + ODConv (4×) | | 35.8 | 57.0 | 38.1 | 20.5 | 38.5 | 45.9 | 31.80M (10.24M) | 176.10G (24.89G) |

settings for all pre-trained models built with CondConv, DyConv and our ODConv, respectively. Experimental details are described in the Appendix.

**Results Comparison.** From the results shown in Table 4, we can observe similar performance improvement trends as on the ImageNet dataset. For Faster R-CNN|Mask R-CNN with the pre-trained ResNet50 backbone models, CondConv (8×) and DyConv (4×) show an AP improvement of 0.9%|0.8% and 1.1%|1.2% to the baseline model respectively, while our method performs much better, e.g., ODConv (1×) with one single convolutional kernel even shows an AP improvement of 1.8%|1.9%, respectively. With the pre-trained MobileNetV2 (1.0×) backbone models, our ODConv (1×) performs obviously better than CondConv (8×), and its performance is on par with that of DyConv (4×) as their AP gap is only 0.2% for both two detectors, striking a better model accuracy and efficiency tradeoff. Similar boosts to AP scores for small, medium and large objects are also obtained by three methods on both detectors. ODConv (4×) always achieves the best AP scores.

## 4.3 ABLATION STUDIES

Finally, we conduct a lot of ablative experiments on the ImageNet dataset, in order to have a better analysis of our ODConv.

Table 5: Results comparison of the ResNet18 models based on ODConv with different settings of the reduction ratio $r$. All models are trained on the ImageNet dataset. Best results are bolded.

| Models | $r$ | Params | MAdds | Top-1 Acc (%) | Top-5 Acc (%) |
|---|---|---|---|---|---|
| ResNet18 | - | 11.69M | 1.814G | 70.25 | 89.38 |
| | 1/4 | 12.58M | 1.839G | **73.41** | **91.29** |
| + ODConv (1×) | 1/8 | 12.15M | 1.838G | 73.11 | 91.10 |
| | 1/16 | 11.94M | 1.838G | 73.10 | 91.10 |

Table 6: Results comparison of the ResNet18 models based on ODConv with different numbers of convolutional kernels $n$. All models are trained on the ImageNet dataset. Best results are bolded.

| Models | $n$ | Params | MAdds | Top-1 Acc (%) | Top-5 Acc (%) |
|---|---|---|---|---|---|
| ResNet18 | - | 11.69M | 1.814G | 70.25 | 89.38 |
| | 1× | 11.94M | 1.838G | 73.10 | 91.10 |
| | 2× | 22.93M | 1.872G | 73.59 | 91.08 |
| + ODConv (r = 1/16) | 3× | 33.92M | 1.894G | 73.77 | 91.35 |
| | 4× | 44.90M | 1.916G | 73.97 | 91.35 |
| | 8× | 88.84M | 2.006G | **74.08** | **91.44** |

**Reduction Ratio Selection.** Our first set of ablative experiments is for the selection of the reduction ratio $r$ used in the attention structure $\pi_i(x)$. From the results shown in Table 5, we can find that under $r = 1/4$, $r = 1/8$ and $r = 1/16$, ODConv consistently obtains large performance improvements to the baseline ResNet18 model (2.85~3.16% top-1 gain), and the extra MAdds are negligible. Comparatively, ODConv with $r = 1/16$ strikes the best tradeoff between model accuracy and efficiency, and thus we choose it as our default setting.

Table 7: Investigating the complementarity of four types of attentions proposed in ODConv. In the experiments, when $\alpha_{wi}$ is used, we set $n = 4$, and otherwise $n = 1$. For the optimal comparison, we use the best $r$ setting reported in Table 5. All ResNet18 models are trained on the ImageNet dataset. Best results are bolded.

| Models | $\alpha_{si}$ | $\alpha_{ci}$ | $\alpha_{fi}$ | $\alpha_{wi}$ | Params | MAdds | Top-1 Acc (%) | Top-5 Acc (%) |
|---|---|---|---|---|---|---|---|---|
| ResNet18 | - | - | - | - | 11.69M | 1.814G | 70.25 | 89.38 |
| | ✓ | - | - | - | 11.98M | 1.827G | 72.42 | 90.76 |
| | - | ✓ | - | - | 12.26M | 1.827G | 72.07 | 90.59 |
| | - | - | ✓ | - | 12.28M | 1.827G | 71.46 | 90.43 |
| + ODConv ($r = 1/4$) | ✓ | ✓ | - | - | 12.27M | 1.827G | 73.13 | 91.14 |
| | ✓ | - | ✓ | - | 12.29M | 1.827G | 72.80 | 90.99 |
| | - | ✓ | ✓ | - | 12.57M | 1.829G | 72.20 | 90.67 |
| | ✓ | ✓ | ✓ | - | 12.58M | 1.839G | 73.41 | 91.29 |
| | ✓ | ✓ | ✓ | ✓ | 45.54M | 1.953G | **74.33** | **91.53** |

Table 8: Comparison of the inference speed (frames per second) for different dynamic convolution methods. All pre-trained models are tested on an NVIDIA TITAN X GPU (with batch size 200) and a single core of Intel E5-2683 v3 CPU (with batch size 1) separately, and the input image size is $224 \times 224$ pixels.

| ResNet50 | Speed on GPU | Speed on CPU | MobileNetV2 (1.0×) | Speed on GPU | Speed on CPU |
|---|---|---|---|---|---|
| Baseline model | 652.2 | 6.4 | Baseline model | 1452.3 | 17.3 |
| + CondConv | 425.1 | 4.0 | + CondConv | 1076.9 | 14.8 |
| + DyConv | 326.6 | 3.8 | + DyConv | 918.2 | 11.9 |
| + DCD | 400.0 | 3.6 | + DCD | 875.7 | 9.5 |
| + ODConv (1×) | 293.9 | 3.9 | + ODConv (1×) | 1029.2 | 12.8 |
| + ODConv (4×) | 152.7 | 2.5 | + ODConv (4×) | 608.3 | 11.2 |

**Convolutional Kernel Number.** Accordingly to the main experiments discussed beforehand, OD-Conv can strike a better tradeoff between model accuracy and size, compared to existing dynamic convolution methods. For a better understanding of this advantage, we next perform the second set of ablative experiments on the ImageNet dataset, training the ResNet18 backbone based on ODConv with different settings of the convolutional kernel number $n$. Table 6 shows the results. It can be seen that ODConv with one single convolutional kernel brings near $3.0\%$ top-1 gain to the ResNet18 baseline, and the gain tends to be saturated when the number of convolutional kernels is set to $8$.

**Four Types of Attentions.** Note that our ODConv has four types of convolutional kernel attentions $\alpha_{si}, \alpha_{ci}, \alpha_{fi}$ and $\alpha_{wi}$ computed along all four dimensions of the kernel space, respectively. Then, we perform another set of ablative experiments on the ResNet18 backbone with different combinations of them to investigate their dependencies. Results are summarized in Table 7, from which the strong complementarity of four types of attentions $\alpha_{si}, \alpha_{ci}, \alpha_{fi}$ and $\alpha_{wi}$ can be clearly observed.

**Inference Speed.** Besides the size and the MAdds for a CNN model, the inference speed at runtime is very important in the practical model deployment. Table 8 provides a comparison of the inference speed for different dynamic convolution methods (both on a GPU and a CPU). It can be seen that the models trained with our ODConv (1×) are faster than the counterparts trained with DyConv and DCD on a CPU. Comparatively, the models trained with CondConv show the fastest run-time speed both on a GPU and a CPU, this is because CondConv is merely added to the final several blocks and the last fully connected layer as discussed in the Method section. When adding ODConv (1×) to the same layer locations as for CondConv, we can obtain more accurate and efficient models.

**More Ablations.** In the Appendix, we provide more ablative experiments to study: (1) the effects of applying ODConv to different layer locations; (2) the importance of the temperature annealing strategy; (3) the choices of the activation functions for $\pi_i(x)$; (4) the influence of the attention sharing strategy; (5) the stability of the model training process; (6) other potentials of ODConv.

## 5 CONCLUSION

In this paper, we present a new dynamic convolution design called Omni-dimensional Dynamic Convolution (ODConv) to promote the representation power of deep CNNs. ODConv leverages a multi-dimensional attention mechanism to learn four types of attentions for convolutional kernels along all four dimensions of the kernel space in a parallel manner, and progressively applying these attentions to the corresponding convolutional kernels can substantially strengthen the feature extraction ability of basic convolution operations of a CNN. Experimental results on the ImageNet and MS-COCO datasets with various prevailing CNN architectures validate its superior performance.

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

# A  APPENDIX

In this section, we describe the supplementary materials including: (1) a computational cost analysis of ODConv; (2) implementation details for the experiments on the ImageNet and MS-COCO datasets; (3) more ablative experiments on the ImageNet dataset; (4) illustrative training and validation curves to compare the stability of the model training process with different dynamic convolution methods; (5) more experiments to study other potentials of ODConv.

## A.1  COMPUTATIONAL COST OF ODCONV

For a convolutional layer with ODConv, its computational cost can be easily calculated according to the design and the multiplication process of four types of attentions shown in Fig. 1 and Fig. 2, respectively. Specifically, following the notations defined in the Method section of the main paper, the extra MAdds of ODConv ($1\times$) over regular convolution (which has $hwk^2c_{in}c_{out}$ MAdds, without consideration of the bias term) can be calculated as

$$MAdds = hwc_{in} + \frac{c_{in}(2c_{in} + c_{out} + k^2)}{r} + k^2c_{in}(1 + 2c_{out}). \tag{3}$$

For ODConv ($n\times$), the extra MAdds over regular convolution can be calculated as

$$MAdds = hwc_{in} + \frac{c_{in}(2c_{in} + c_{out} + k^2 + n)}{r} + k^2c_{in}(1 + c_{out} + 2nc_{out}). \tag{4}$$

Compared to $hwk^2c_{in}c_{out}$ MAdds for regular convolution, the extra MAdds by ODConv are small.

## A.2  EXPERIMENTAL DETAILS

**Experimental Details on ImageNet.** Recall that we use MobileNetV2 (Sandler et al., 2018) and ResNet (He et al., 2016) families for experiments on the ImageNet dataset, covering both lightweight CNN architectures and larger ones. For fair comparisons, we adopt the popular training settings used in the community to train the respective backbone models with all methods. Specifically, for ResNet18, ResNet50 and ResNet101, all models are trained with SGD for 100 epochs. We set the batch size as 256, the weight decay as 0.0001 and the momentum as 0.9. The learning rate starts at 0.1, and is divided by 10 every 30 epochs. Following DyConv (Chen et al., 2020), for our ODConv, we also use dropout rate of 0.1 for ResNet18. We use dropout rate of 0.2 for ResNet50 and ResNet101 for our ODConv. For MobileNetV2 ($1.0\times, 0.75\times, 0.5\times$), all models are trained with SGD for 150 epochs (we also have the experiments for training all models with 300 epochs in this Appendix). We set the batch size as 256, the weight decay as 0.00004 and the momentum as 0.9. The learning rate starts at 0.05, and is scheduled to arrive at zero within a single cosine cycle. Following DyConv, for our ODConv, we also use dropout rate of 0.2 for MobileNetV2 ($1.0\times$) and dropout rate of (0.1, 0) for MobileNetV2 ($0.75\times, 0.5\times$). Regarding the temperature annealing strategy used for DyConv and ODConv, the temperature reduces from 30 to 1 linearly in the first 10 epochs for all models. All experiments are performed on the servers having 8 GPUs. We follow the standard protocols to train and evaluate each CNN backbone. For training, images are resized to $256 \times 256$ first, and then $224 \times 224$ crops are randomly sampled from the resized images or their horizontal flips normalized with the per-channel mean and standard deviation values. For evaluation, we report top-1 and top-5 recognition rates using the center image crops.

**Experimental Details on MS-COCO.** Recall that we use the popular MMDetection toolbox (Chen et al., 2019) for experiments on the MS-COCO dataset with the pre-trained ResNet50 and MobileNetV2 ($1.0\times$) models as the backbones for the detector. We select the mainstream Faster R-CNN (Ren et al., 2015) and Mask R-CNN (He et al., 2017) detectors with Feature Pyramid Networks (FPNs) (Lin et al., 2017) as the necks to build the basic object detection systems. For a neat and fair comparison, the convolutional layers in the FPN necks still use regular convolutions, and we maintain the same data preparation pipeline and hyperparameter settings for all pre-trained models built with CondConv, DyConv and our ODConv, respectively. All experiments are performed on a server having 8 GPUs with a mini-batch size of 2 images per GPU. We finetune these detectors on the MS-COCO training set following the $1\times$ learning rate schedule, which indicates a total of 12 epochs with the learning rate divided by 10 at the $8^{th}$ epoch and the $11^{th}$ epoch, respectively.

Following DyConv, the temperature annealing strategy is not used in the downstream experiments on the MS-COCO dataset, as it is prone to worse results. In the validation, we report the standard Average Precision (AP) under IOU thresholds ranging from 0.5 to 0.95 with an increment of 0.05. We also keep AP scores for small, medium and large objects.

### A.3   MORE ABLATIVE EXPERIMENTS ON IMAGENET

Besides the ablative experiments described in the main paper, here we provide more ablative experiments, for a better understanding of our ODConv.

**Effects of Applying ODConv to Different Layer Locations.**   In the Experiments section of the main paper, we show that applying dynamic convolution to less layers leads to a faster runtime speed compared to applying it to more layers. For easy implementation, we follow DyConv and apply ODConv to all convolutional layers except the first layer of each CNN architecture tested in our main experiments. However, it is necessary to study the effects of applying ODConv to different layer locations. The performance comparison of adding ODConv to different layers of the MobileNetV2 $(0.5\times)$ backbone is given in Table 9, showing that our default setting (i.e., adding ODConv to all convolutional layers except the first layer) has the largest gain. This indicates that adding ODConv to more convolutional layers tends to have a model with higher recognition accuracy.

Table 9: Results comparison of adding ODConv to different layers of the MobileNetV2 $(0.5\times)$ backbone. All models are trained on the ImageNet dataset for 150 epochs, and we set $r = 1/16$ and $n = 1$. For the inference speed, all pre-trained models are tested on an NVIDIA TITAN X GPU (with batch size 200) and a single core of Intel E5-2683 v3 CPU (with batch size 1) separately, and the input image size is $224 \times 224$ pixels. Best results are bolded.

| Models | Params | MAdds | Top-1 Acc (%) | Top-5 Acc (%) | Speed on GPU | Speed on CPU |
|---|---|---|---|---|---|---|
| MobileNetV2 $(0.5\times)$ | 2.00M | 97.1M | 64.30 | 85.21 | 2840.9 | 30.1 |
| + ODConv (to all conv layers except the first layer) | 2.43M | 101.8M | **68.26** | **87.98** | 1837.4 | 18.3 |
| + ODConv (to all 1×1 conv layers) | 2.26M | 100.2M | 67.43 | 87.41 | 2083.3 | 27.1 |
| + ODConv (to all 3×3 conv layers) | 2.17M | 99.1M | 67.54 | 87.55 | 2064.9 | 27.4 |
| + ODConv (to last 6 residual blocks) | 2.31M | 99.1M | 67.29 | 87.39 | 1879.5 | 19.6 |

**Importance of Temperature Annealing Strategy.**   Following DyConv, we also use the temperature annealing strategy to facilitate the model training process of our ODConv. To study its effect, we perform a set of ablative experiments on the ImageNet dataset, training the ResNet18 backbone based on ODConv with or without using the temperature annealing strategy. From the results shown in Table 10, we can find that the temperature annealing strategy is really important, which brings $1.06\%|1.04\%$ top-1 improvement to the ResNet18 model based on ODConv $(1\times)$|ODConv $(4\times)$.

Table 10: Analyzing the effect of the temperature annealing when training ResNet18 with ODConv. All models are trained on the ImageNet dataset, and we set $r = 1/4$. Best results are bolded.

| Models | Temperature Annealing | Top-1 Acc (%) | Top-5 Acc (%) |
|---|---|---|---|
| ResNet18 | - | 70.25 | 89.38 |
| + ODConv $(1\times)$ | ✓ | **73.41** | **91.29** |
| | - | 72.35 | 90.65 |
| + ODConv $(4\times)$ | ✓ | **74.33** | **91.53** |
| | - | 73.29 | 90.95 |

**Choices of the Activation Functions.**   Recall that in the structure of $\pi_i(x)$ for our ODConv, we use two different activation functions (either a Sigmoid function or a Softmax function) to compute four types of attentions $\alpha_{si}$, $\alpha_{ci}$, $\alpha_{fi}$ and $\alpha_{wi}$. It is also critical to compare the performance of different activation functions for ODConv. Since the activation function choices for the channel dimension and the convolutional kernel dimension have been thoroughly discussed in the papers of SE and DyConv respectively, we adopt their suggestions for three related attentions defined in our ODConv. Here we perform another set of ablative experiments, focusing on the choices of the activation function for the spatial dimension. With the ImageNet dataset, we use ResNet18 as the test backbone network to explore the effects of different activation functions for computing the attention of ODConv along the spatial dimension of the kernel space. Table 11 summarizes the results, from which we can see that the Sigmoid function performs better than the Softmax function, and thus we use the Sigmoid function to compute the attention scalars along the spatial dimension for ODConv.

**Attention Sharing Strategy.**   As we discussed in the Method section, we share three attentions $\alpha_{si}$, $\alpha_{ci}$ and $\alpha_{fi}$ to all convolutional kernels for the easy implementation as well as for more efficient

Table 11: Results comparison of ODConv with different activation functions. All models are trained on the ImageNet dataset, and we set $r = 1/4$. Best results are bolded.

| Models | Activation Function | Top-1 Acc (%) | Top-5 Acc (%) |
|---|---|---|---|
| ResNet18 | - | 70.25 | 89.38 |
| + ODConv (1×) | Sigmoid | **73.41** | **91.29** |
| | Softmax | 73.23 | 91.19 |
| + ODConv (4×) | Sigmoid | **74.33** | **91.53** |
| | Softmax | 73.97 | 91.50 |

Table 12: Results comparison of ODConv with or without using the attention sharing strategy. All models are trained on the ImageNet dataset. Best results are bolded.

| Models | Params | MAdds | Top-1 Acc (%) | Top-5 Acc (%) |
|---|---|---|---|---|
| ResNet18 | 11.69M | 1.814G | 70.25 | 89.38 |
| + ODConv w/ attention sharing ($r = 1/16, 4×$) | 44.90M | 1.916G | 73.97 | 91.35 |
| + ODConv w/o attention sharing ($r = 1/16, 4×$) | 45.43M | 1.950G | **74.16** | **91.47** |

training. Actually, training separate $\alpha_{si}$, $\alpha_{ci}$ and $\alpha_{fi}$ for each additive convolutional kernel will bring further improved model accuracy, as can be seen from the ablative results in Table 12. That is, when preferring to get more accurate models, the attention sharing strategy would be removed.

**Training MobileNetV2 Backbones for 300 Epochs.** Recall that in the Experiments section of the main paper, we compare our ODConv with CondConv, DyConv and DCD on MobileNetV2 (1.0×, 0.75×, 0.5×), following the popular training settings used in the community, where all models are trained for 150 epochs. To validate the effectiveness of our ODConv further, we also conduct another set of experiments using an increased number of training epochs (300 instead of 150) as adopted in DyConv (Chen et al., 2020). All the other training settings remain the same to those for the experiments described in the main paper. For a clean comparison of all methods, we do not use advanced training tricks such as mixup (Zhang et al., 2018a) and label smoothing (Szegedy et al., 2016). Results are summarized in Table 13 where performance gains show a similar trend as those in Table 1 for the model training with 150 epochs. Results in Fig. 3 further show that our ODConv gets a better trade-off between model accuracy and size even on the light-weight MobileNetV2 backbones compared to CondConv and DyConv.

Table 13: Results comparison on the ImageNet validation set with MobileNetV2 (1.0×, 0.75×, 0.5×) as the backbones. All models are trained for 300 epochs. Best results are bolded.

| Models | Params | MAdds | Top-1 Acc (%) |
|---|---|---|---|
| MobileNetV2 (1.0×) | 3.50M | 300.8M | 72.07 |
| + CondConv (8×) | 22.88M | 318.1M | 74.31(↑2.24) |
| + DyConv (4×) | 12.40M | 317.1M | 75.08(↑3.01) |
| + DCD | 5.72M | 318.4M | 74.48(↑2.41) |
| + ODConv (1×) | 4.94M | 311.8M | 75.13(↑3.06) |
| + ODConv (4×) | 11.52M | 327.1M | **75.68(↑3.61)** |
| MobileNetV2 (0.75×) | 2.64M | 209.1M | 69.76 |
| + CondConv (8×) | 17.51M | 223.9M | 72.18(↑2.42) |
| + DyConv (4×) | 7.95M | 220.1M | 73.48(↑3.72) |
| + DCD | 4.08M | 222.9M | 72.39(↑2.63) |
| + ODConv (1×) | 3.51M | 217.1M | 73.03(↑3.27) |
| + ODConv (4×) | 7.50M | 226.3M | **74.45(↑4.69)** |
| MobileNetV2 (0.5×) | 2.00M | 97.1M | 65.10 |
| + CondConv (8×) | 13.61M | 110.0M | 68.45(↑3.35) |
| + DyConv (4×) | 4.57M | 103.2M | 69.83(↑4.73) |
| + DCD | 3.06M | 105.6M | 70.08(↑4.98) |
| + ODConv (1×) | 2.43M | 101.8M | 69.16(↑4.06) |
| + ODConv (4×) | 4.44M | 106.4M | **70.87(↑5.77)** |

A.4    ILLUSTRATION OF MODEL TRAINING AND VALIDATION CURVES

Fig. 4 illustrates the training and validation accuracy curves of the ResNet18 models trained on the ImageNet dataset with CondConv, DyConv, ODConv (1×) and ODConv (4×), respectively. We can see that our ODConv shows consistent high top-1 gains throughout the training process compared to the other three dynamic convolution counterparts.

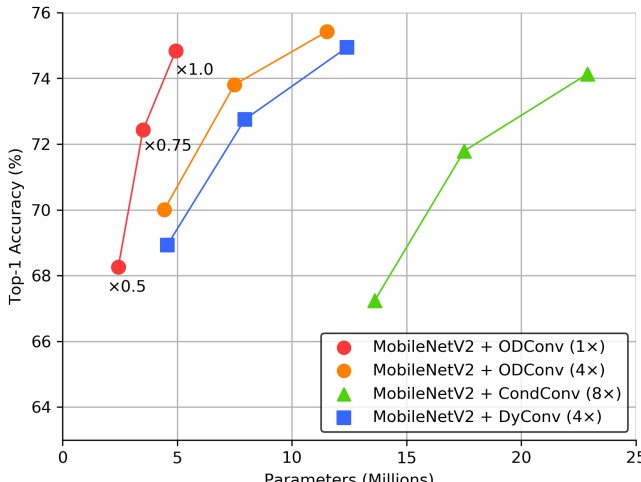

Figure 3: Comparison of model accuracy and size for the pre-trained MobileNetV2 models based on different dynamic convolution methods. All models are trained for 150 epochs on the ImageNet dataset. It can be seen that our ODConv (1×) makes a better accuracy and size tradeoff for the light-weight MobileNetV2 backbones compared to CondConv and DyConv. On the larger ResNet18 and ResNet50 backbones, even better results are obtained by ODConv, as can be seen from the results shown in Table 2 of the main paper.

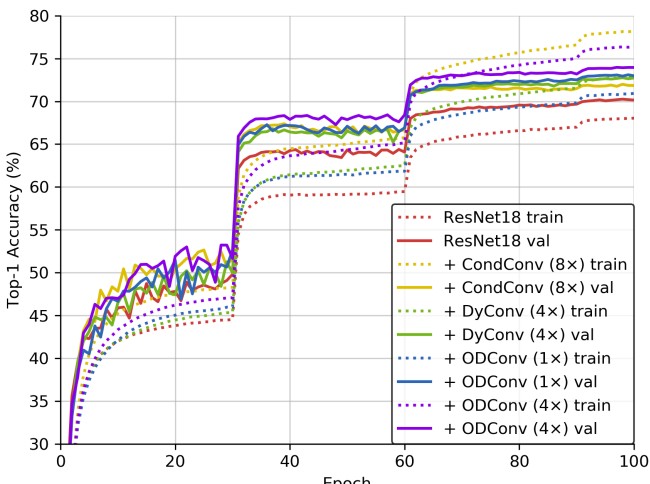

Figure 4: Curves of top-1 training accuracy (dashed line) and validation accuracy (solid line) of the ResNet18 models trained on the ImageNet dataset with CondConv (8×), DyConv (4×), our ODConv (1×) and ODConv (4×), respectively. Comparatively, our ODConv (1×) outperforms both CondConv and DyConv yet has only 14.68%|26.26% parameters of the model trained with CondConv|DyConv. Our ODConv (4×) converges with the best validation accuracy, which outperforms CondConv|DyConv by 1.98%|1.21% top-1 gain with fewer parameters.

## A.5 MORE EXPERIMENTS FOR STUDYING OTHER POTENTIALS OF ODCONV

In this section, we provide a lot of extra experiments conducted for studying other potentials of ODConv.

**Performance Comparison on A CNN Backbone Added with the SE Module.** Recall that SE (Hu et al., 2018b) performs attention on the output features of a convolutional layer, while our ODConv performs attention on the convolutional kernels, and thus they are likely to be complimentary. To validate this, we perform another set of experiments on the ImageNet dataset with ResNet18 backbone. In the experiments, we use the ResNet18+SE variant reported in Table 2 as the backbone, and train this backbone with CondConv, DyConv, DCD and our ODConv separately, adopting the same training settings used for the ResNet18 backbone. Results are shown in Table 14. It can be seen that our ODConv brings significantly large performance improvement to the ResNet18 backbone already incorporating the SE module, showing its great combination potential.

Table 14: Results comparison on the ImageNet validation set with the ResNet18+SE backbone trained for 100 epochs. For our ODConv, we set $r = 1/16$. Best results are bolded.

| Models | Params | MAdds | Top-1 Acc (%) | Top-5 Acc (%) |
|---|---|---|---|---|
| ResNet18 | 11.69M | 1.814G | 70.25 | 89.38 |
| + SE | 11.78G | 1.816G | 70.98 (↑0.73) | 90.03 (↑0.65) |
| + SE & ODConv(4×) | 44.99M | 1.918G | **74.04 (↑3.79)** | **91.38 (↑2.00)** |

**Performance Boost Using Heavy Data Augmentations and a Longer Training Schedule.** Recall that in the main paper, we do not use advanced training tricks such as mixup (Zhang et al., 2018a) and label smoothing (Szegedy et al., 2016), aiming to have clean performance comparisons. To better explore the potential of our ODConv, it would be necessary to have a study about whether ODConv can also work well when using heavy data augmentations and a longer training schedule. To this end, we perform a set of experiments on the ImageNet dataset with the ResNet18 backbone trained with ODConv (4×). In the experiments, we set $r = 1/16$ for our method, first train ResNet18 with label smoothing, mixup and (label smoothing+mixup) separately for 100 epochs (the other training settings are the same to those used for Table 2), and then train ResNet18 with (label smoothing+mixup) for 120 epochs instead of 100 epochs. Detailed results are summarized in Table 15. It can be seen that our ODConv works well when using aggressive data augmentations and a longer training schedule, showing further improved performance.

Table 15: Results comparison of the ResNet18 models based on ODConv with aggressive data augmentations and a longer training schedule. All models are trained on the ImageNet dataset. Best results are bolded.

| Models | Top-1 Acc (%) | Top-5 Acc (%) |
|---|---|---|
| ResNet18 + ODConv ($r = 1/16$, 4×) | 73.97 | 91.35 |
| + Label Smoothing | 74.15 (↑0.18) | 91.42 (↑0.07) |
| + Mixup | 74.05 (↑0.08) | 91.63 (↑0.28) |
| + Label Smoothing & Mixup | 74.18 (↑0.21) | 91.68 (↑0.33) |
| + Label Smoothing & Mixup & Longer Training Schedule | **74.59 (↑0.62)** | **91.74 (↑0.39)** |

**Feature Pooling Strategy.** Note that when computing four types of attentions in our ODConv, the inputs features are always globally averaged, reducing their spatial dimension to one pixel. According to the first three results of Table 7 in the main paper, the spatial convolutional kernel attention $\alpha_{si}$ brings the largest gain to the baseline model compared to the other two attentions ($\alpha_{ci}$ and $\alpha_{fi}$ computed along the dimension of the input channels and the output channels, respectively). Intuitively, at the first glance, it may easily lead to a counterintuitive feeling as the input features used to predict $\alpha_{si}$ (as well as the other three attentions) do not include spatial information after the global average pooling. However, it should be noted that our ODConv uses the attentions generated from the input features to modify the convolutional kernels, which is quite different from popular self-calibration attention mechanisms (e.g., SE (Hu et al., 2018b) and CBAM (Woo et al., 2018) use the channel attentions generated from the output features of a convolutional layer to recalibrate the output features themselves). As our ODConv is not a self-calibration attention module, reducing the spatial dimension of the input features to one pixel (but the length of the reduced vector is large enough) will not affect its promising performance. To validate this, we conduct another experiment on the ImageNet dataset with the ResNet18 backbone. In the experiment, for $\alpha_{si}$, we pool the input features to have a size of 3×3 instead of 1×1, and show the results in Table 16. It can be seen that preserving more spatial information only brings 0.06%|0.02% gain to top-1|top-5 accuracy, which proves the effectiveness of our current design to a large degree.

Table 16: Comparison of the feature pooling strategy in ODConv using different spatial sizes for the reduced features. All ResNet18 models are trained on the ImageNet dataset. Best results are bolded.

| Models | Params | MAdds | Top-1 Acc (%) | Top-5 Acc (%) |
|---|---|---|---|---|
| ResNet18 | 11.69M | 1.814G | 70.25 | 89.38 |
| + ODConv ($r = 1/4$, $1\times$, pooling to $1\times1$) | 12.58M | 1.839G | 73.41 (↑3.16) | 91.29 (↑1.91) |
| + ODConv ($r = 1/4$, $1\times$, pooling to $3\times3$) | 14.85M | 1.842G | **73.47 (↑3.22)** | **91.31 (↑1.93)** |

**Comparison of Training Cost.** Recall that in the main paper, we provide a comparison of the inference speed for different dynamic convolution methods. Here, we further perform a set of experiments to compare the training cost of our ODConv, CondConv, DyConv and DCD. Specifically, experiments are performed on the ImageNet dataset with the ResNet50 and MobileNetV2 backbones, using the same training settings as in the main paper. Regarding the training cost, we report results in terms of three metrics (seconds per batch, minutes per epoch, and the total number of hours for the whole training) in Table 17. Regarding different dynamic convolution methods, it can be seen that the training cost trend is somewhat similar to that for the runtime inference speed reported in Table 8. Generally, the training cost for dynamic convolution methods is obviously heavier (about $2\times$ to $4\times$) than that for the baseline model, which is mainly due to the time-intensive back-propagation process for additive convolutional kernels weighted with input-dependent attentions.

Table 17: Comparison of the training cost for different dynamic convolution methods. All models are trained on the ImageNet dataset using the server with 8 NVIDIA TITAN X GPUs. We report results in terms of three metrics (seconds per batch, minutes per epoch, and the total number of hours for the whole training).

| Network | ResNet50 | | | MobileNetV2 (1.0×) | | |
|---|---|---|---|---|---|---|
| Models | Batch Cost (second) | Epoch Cost (minute) | Total Cost (hour) | Batch Cost (second) | Epoch Cost (minute) | Total Cost (hour) |
| Baseline | 0.216 | 18.4 | 31.7 | 0.113 | 9.7 | 25.3 |
| + CondConv (8×) | 0.429 | 36.4 | 62.8 | 0.175 | 15.1 | 39.3 |
| + DyConv (4×) | 0.569 | 48.0 | 82.3 | 0.310 | 26.4 | 67.9 |
| + DCD | 0.414 | 35.1 | 60.4 | 0.282 | 24.0 | 61.9 |
| + ODConv (1×) | 0.513 | 43.4 | 73.5 | 0.301 | 25.6 | 65.7 |
| + ODConv (4×) | 0.896 | 75.5 | 129.3 | 0.402 | 34.0 | 86.8 |

**A Deep Understanding of Learnt Attention Values.** Note that our ODConv leverages a multi-dimensional attention mechanism with a parallel strategy to learn four types of attentions $\alpha_{si}$, $\alpha_{ci}$, $\alpha_{fi}$, $\alpha_{wi}$ for convolutional kernels along all four dimensions of the kernel space at any convolutional layer. In principle, progressively multiplying these four types of attentions to the convolutional kernel in the location-wise, channel-wise, filter-wise and kernel-wise orders makes convolution operations be different w.r.t. all spatial locations, all input channels, all filters and all kernels for each input sample, providing a performance guarantee to capture rich context cues. To have a better understanding of this powerful attention mechanism, it is necessary to study the learnt attention values. To this end, we use the well-trained ResNet18 models reported in Table 7 and all 50,000 images in the ImageNet validation set, and conduct a set of experiments to analyze the learnt attention values for $\alpha_{si}$, $\alpha_{ci}$, $\alpha_{fi}$, $\alpha_{wi}$ and their full combination by providing lots of visualization examples (obtained with the Grad-CAM++ method (Chattopadhyay et al., 2018)) and the statistical distributions across different layers. The detailed computational process and results are shown in Fig. 5 and Fig. 6. We can get the following observations: (1) Four types of attentions $\alpha_{si}$, $\alpha_{ci}$, $\alpha_{fi}$ and $\alpha_{wi}$ are complementary to each other by visualization examples, which echoes the conclusion from Table 7. The most salient attention will combat the failure cases of the other three attentions, helping the model to produce accurate predications; (2) All four types of attentions demonstrate varying attention value distribution trends across different layers of the trained model, showing their capability to capture rich context cues; (3) Each attention has its own value distribution trend, which is more diverse for $\alpha_{si}$ and $\alpha_{wi}$ than for $\alpha_{ci}$ and $\alpha_{fi}$, showing $\alpha_{si}$ and $\alpha_{wi}$ maybe more important to some degree. Note that the dimension of attentions $\alpha_{si}$, $\alpha_{ci}$, $\alpha_{fi}$ and $\alpha_{wi}$ is quite different, e.g., 9 for $\alpha_{si}$ with $3 \times 3$ convolutional kernels, 4 for $\alpha_{wi}$ with $n = 4$, and typically several hundred for $\alpha_{ci}$ and $\alpha_{fi}$. This is another perspective why they are different and complementary in their design nature regarding all four dimensions of the convolutional kernel space for any convolutional layer.

**Limitations of ODConv.** On the one side, according to the results shown in Table 1, Table 2 and Table 8, although our ODConv shows obviously better performance under the similar model size

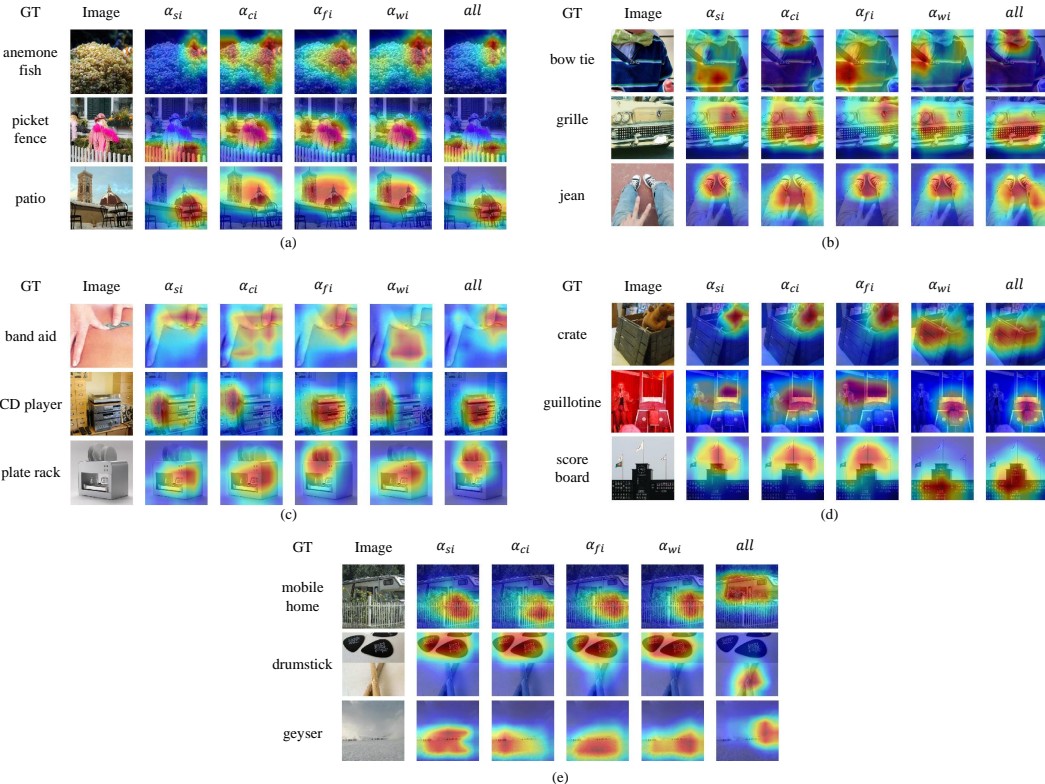

Figure 5: Comparison of illustrative visualization results with Grad-CAM++ (Chattopadhyay et al., 2018). Results are obtained from the pre-trained ResNet18 models (reported in Table 7) with $\alpha_{si}$, $\alpha_{ci}$, $\alpha_{fi}$, $\alpha_{wi}$, and all of them (i.e., ODConv (4×)), separately. (a) The model with $\alpha_{si}$ and the model with all four attentions make the right predications, while the other three models with one single attention fail. (b) The model with $\alpha_{ci}$ and the model with all four attentions make the right predications, while the other three models with one single attention fail. (c) The model with $\alpha_{fi}$ and the model with all four attentions make the right predications, while the other three models with one single attention fail. (d) The model with $\alpha_{wi}$ and the model with all four attentions make the right predications, while the other three models with one single attention fail. (e) Only the model with all four attentions makes the right predications, while the other four models with one single attention fail. These visualization results further backup the conclusion observed from Table 7. Best viewed with zoom-in.

(e.g., ODConv (4×) vs. DyConv (4×)), it leads to slightly increased FLOPs and introduces extra latency to the runtime inference speed, as the proposed four types of convolutional kernel attentions $\alpha_{si}$, $\alpha_{ci}$, $\alpha_{fi}$ and $\alpha_{wi}$ introduce a bit more learnable parameters. Under the similar model size, the training cost of our ODConv is also heavier than reference methods (e.g., ODConv (4×) vs. DyConv (4×)), according to the results shown in Table 17. On the other side, although we provide a wide range of ablative studies to analyze the effect of different hyperparameters of our ODConv using the ResNet18 backbone with the ImageNet dataset, and apply the resulting combination of the hyperparameters to all backbone networks, it is not the optimal setting to different backbone networks. This suggests the way to reduce the computational cost of our method, finding a proper combination to constrain the final model to the target speed and accuracy for a particular circumstance. Besides, the potential of applying ODConv to more deep and large backbones beyond ResNet101 has not been explored due to the constraint of our available computational resource.

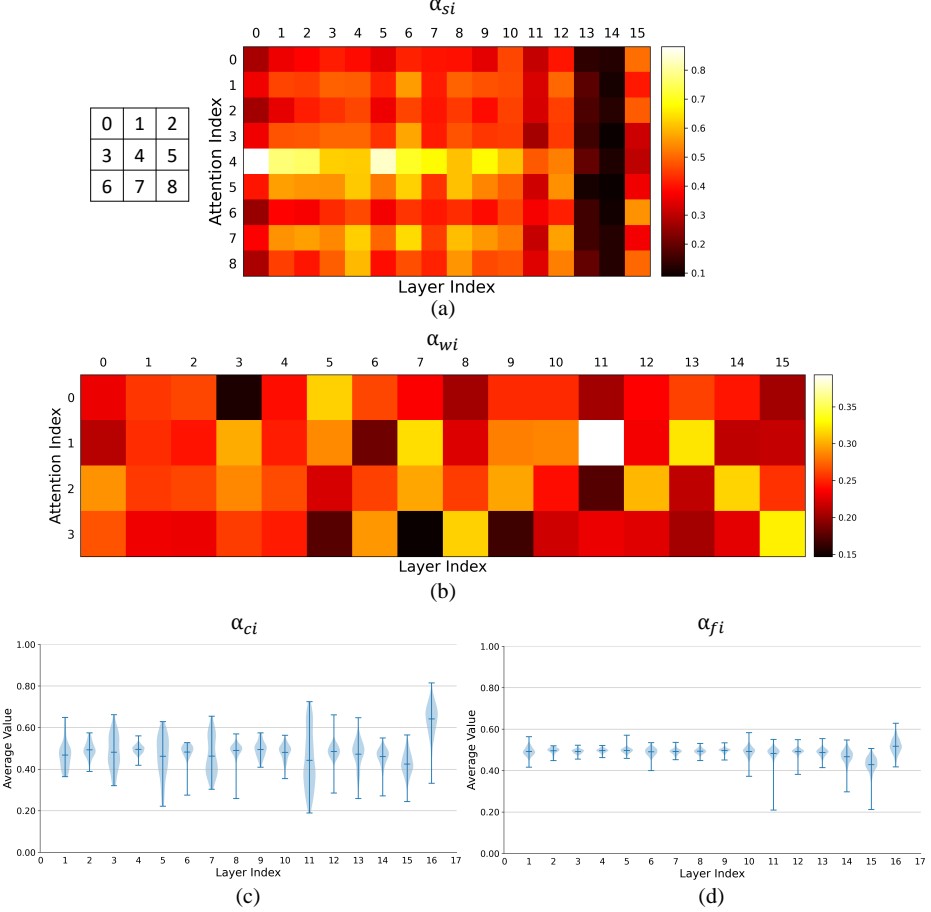

Figure 6: Comparison of the statistical distributions of the learnt attention values regarding $\alpha_{si}$, $\alpha_{ci}$, $\alpha_{fi}$ and $\alpha_{wi}$ across different layers of the pre-trained ResNet18 model with ODConv ($4\times$). We run the model on all 50,000 images from the ImageNet validation dataset to first collect learnt attention values for $\alpha_{si}$, $\alpha_{ci}$, $\alpha_{fi}$ and $\alpha_{wi}$ separately, and then compute the layer-wise statistical distribution for each of them over all image samples. (a) For $\alpha_{si}$, we show the mean attention value for each weight of $3\times3$ convolutional kernels. (b) For $\alpha_{wi}$, we show the mean attention value for each of 4 convolutional kernels. (c) For $\alpha_{ci}$, we show the statistical distribution of the mean attention value over all input channels corresponding to 4 convolutional kernels. (d) For $\alpha_{fi}$, we show the statistical distribution of the mean attention value over all convolutional filters corresponding to 4 convolutional kernels. Best viewed with zoom-in.

