# OpenReview forum: "Omni-Dimensional Dynamic Convolution"
_ICLR.cc/2022/Conference — ICLR 2022 Spotlight_

### Official Review · Reviewer_PSFr · 2021-10-21

**Correctness:** 4
**Technical Novelty And Significance:** 4
**Empirical Novelty And Significance:** 1
**Recommendation:** 8
**Confidence:** 5

**Main Review:**

The paper starts by analyzing the problem of traditional dynamic convolution frameworks, e.g. CondCond and DyConv, and find the previous networks are not fully adapted with the kernel-wise dynamic coefficient. In order to overcome this issue, the author proposed to dynamically weight all four dimensions for the convolutional kernel. This idea can be viewed as a more general form of dynamic convolution and SE, which I think is quite novel. Another strength of this paper is its comprehensive experiments. Both the results in the main paper and the supplementary further justify the benefits brought by the newly designed ODConv.

My main concerns about this submission are as follows.

1. In Table 7, the ablations for the performance of dynamic operation added on different dimensions are shown. From the first three results, I observe the gain brought by the dynamics added on the spatial dimension (70.25% vs 72.42%) is the largest compared to the other two, which is very weird to me. Based on the design of the branch that generates dynamic coefficients, I find the features are always globally averaged, which removes the spatial information. I can hardly understand why the spatially averaged features can still generate the dynamic coefficients to adjust the kernels from the spatial view. Furthermore, I doubt the effectiveness of adding dynamics to the input channels, since the input features have been dynamically adapted by the previous layer. Although non-linearity might exist between two convolutional layers, the activation functions do not fuse channels and will not thoroughly break the dynamic properties. Thus I doubt some offsets might happen and weaken the efficiency of the proposed model.

2. The experimental results are just built on ResNet and MobileNet V2. MobileNet V3, which is more efficient, is not compared with. I wonder whether the author can further compare ODConv to CondConv and DCD with MobileNet V3 as backbone. Moreover, in the object detection part, I do not find the results for DCD. I'm curious about how DCD performs on object detection and whether the proposed ODConv is still better than DCD.

3. In Table 8, the inference time is tested, which is good. But I do not find the latency for the static model. Besides, for the inference time tested on CPU, I'm not sure whether multiple cores are used or not. Please clarify.

4. In the paper the author claim "ODConv with only filter-wise attention αf1 will be reduced into performing SE" is not accurate. For SE, the expression is y=f(x) \dot x, while ODConv with only filter-wise attention is y = f(x) \dot wx. The key difference is whether the dynamic coefficient f(x) is generated based on the input or output of the convolutional layers.

**Summary Of The Paper:**

This work mainly focuses on designing a new dynamic network for large-scale image recognition problems. Specifically, the author discussed the weakness of the existing dynamic convolution operations, and based on the analysis the author proposed a novel framework with the name ODConv. Extensive experiments confirm the superiority of the proposed new framework.

**Summary Of The Review:**

Overall speaking, the paper has novelty and very comprehensive experiments to support it. I am prone to accept this submission. But the weakness of this paper is too experimental-oriented without thorough analysis on the results. My first concern is critical, since I want to understand what happens when different dimension is dynamically adapted. If possible, I hope the author to add some visualization or other analysis on the dynamic coefficient from the statistical view to help readers have a better understand of the proposed dynamic mechanism.

---

> ### Author Response · Authors · 2021-11-22
> **Responses to Official Review by Reviewer PSFr: Part 2**
>
> 2.**To your second concern** “The experimental results are just built on ResNet and MobileNet V2. MobileNet V3, which is more efficient, is not compared with. I wonder whether the author can further compare ODConv to CondConv and DCD with MobileNet V3 as backbone. Moreover, in the object detection part, I do not find the results for DCD. I'm curious about how DCD performs on object detection and whether the proposed ODConv is still better than DCD.”
>
> **Our responses** are: **(1)** Thanks for your constructive comment. Accordingly, we performed a set of experiments on the ImageNet dataset with the MobileNetV3-Small backbone. In the experiments, we train this backbone network with CondConv, DyConv, DCD and our ODConv separately, adopting the same training settings used for the MobileNetV2 backbones (see page 13 of the revised manuscript for experimental details). Detailed results are summarized in the below Table. Note that the MobileNetV3-Small backbone is constructed with the SE module, even for such a strong light-weight backbone network, our ODConv (4$\times$) obviously performs better than all counterparts, and our ODConv (1$\times$) with the smallest model size also outperforms 3 of 4 counterparts. **(2)** The reason for not including the object detection results of DCD in Table 4 is that object detection is not considered in the original paper of DCD. Following your constructive comment, we test DCD on the MS-COCO dataset using the same settings described in our manuscript, and add the corresponding results to Table 4. It can be seen that the performance of DCD is on par with those for DyConv or CondConv, while our ODConv (4$\times$) and ODConv (1$\times$) perform consistently better than DCD, mostly showing clear margins for two detectors with different pretrained backbone models.
>
> | Models | Params | MAdds | Top-1 Acc (%) | Top-5 Acc (%) |
> | -- | :-------: |  :-------: | :--------: |  :--------: |
> | MobileNetV3-Small | 2.94M | 61.1M | 67.14 | 87.36 |
> | + CondConv (8$\times$) | 14.39M | 74.8M | 68.06 | 87.62 |
> | + DyConv (4$\times$) | 4.77M | 64.7M| 69.13 | 88.31 |
> | + DCD | 3.57M | 65.7M | 68.18 | 87.71 |
> | + ODConv (1$\times$) | 3.29M | 63.7M | 68.45 | 87.96 |
> | + ODConv (4$\times$) | 4.65M | 66.9M | **69.68** | **88.64** |
>
> 3.**To your third concern** “In Table 8, the inference time is tested, which is good. But I do not find the latency for the static model. Besides, for the inference time tested on CPU, I'm not sure whether multiple cores are used or not. Please clarify.”
>
> **Our responses** are: **(1)** Table 8 mainly compares the inference speed of our method and existing dynamic convolution designs (CondConv, DyConv and DCD), so the runtime latency for the static model was not considered. Following your constructive suggestion, we test the inference speed (on a single GPU/a single CPU core) for the static model, add the corresponding results to Table 8. Generally, the static model runs with a faster runtime speed compared to all corresponding models based on dynamic convolutions. **(2)** Yes, for the inference speed on CPU, it is tested using a single CPU core. We revised the caption of Table 8 to make it clearer.
>
> 4.**To your last concern** “In the manuscript the author claim "ODConv with only filter-wise attention αf1 will be reduced into performing SE" is not accurate. For SE, the expression is y=f(x) \dot x, while ODConv with only filter-wise attention is y = f(x) \dot wx. The key difference is whether the dynamic coefficient f(x) is generated based on the input or output of the convolutional layers.
>
> **Our responses** are: **(1)** Yes, in this special case, for a convolutional layer, one key difference of the original SE and our ODConv with only filter-wise attention $\alpha_{f1}$ is that the original SE is conditioned on the output features $y$ while our design is conditioned on the input features $x$. **(2)** Another key difference is that the original SE uses its dynamic attentions $f(\cdot)$ conditioned on the output features $y$ to recalibrate $y$ themselves, while our design uses dynamic attentions $f(\cdot)$ conditioned on the input features $x$ to recalibrate the convolutional kernels $w$. **(3)** Actually, in the original manuscript, we partially clarified this in the sentences subsequent to your mentioned sentence "ODConv with only filter-wise attention $\alpha_{f1}$ will be reduced into performing SE". Following your insightful comment, we revised the related description to make it clearer and accurate.
>
> **Finally**, regarding more experiments and discussions that we have made during the rebuttal phase, you are referred to our top-level comments titled **“The Summary of Our Responses to All Official Reviews”**, our responses to the other reviewers, and the revised manuscript.

---

> > ### Comment · Reviewer_PSFr · 2021-11-22
> > **comments on rebuttal**
> >
> > Thanks for providing such a comprehensive rebuttal. I'm very satisfied and will increase the original rating. The one remaining concern I have is still the dynamics added on the kernel. I can understand that the dynamic coefficients are obtained based on the input features, but it still looks weird to me that a spatially pooled feature can still generate a spatial kernel. If it is possible, I want to hear more feedback from the authors. But since the author has shown that even the features are pooled to 3x3, the performance is not increased, I can just view it as an experimental finding that needs further study and will not influence my overall judgement on the quality of this paper.

---

> > > ### Author Response · Authors · 2021-11-24
> > > **More feedback regarding the spatial kernel attention**
> > >
> > > We are so excited that you well recognized our rebuttal, and sincerely thank you for really constructive comments and the recognition of our work.
> > >
> > > We would be more than happy, try our best, to discuss more on why a spatially pooled feature can still generate a spatial kernel attention. **(1)** As you already know, in our ODConv, the spatial kernel attention $\alpha_{si}$ together with {$\alpha_{ci}$, $\alpha_{fi}$, $\alpha_{wi}$} are all generated based on the input features of a convolutional layer. Under this context, there is no explicit spatial/geometric correspondence between $h\times w$ pixel locations of the input features and $k\times k$ weight locations of the convolutional kernels. Existing works [1-2] for studying feature visualization of pretrained CNN models show that there usually exist some salient pixels in each output feature channel at a convolutional layer, and the locations of salient pixels for different output feature channels are also different, and these phenomena vary across different layers. Note that the output features of the current convolutional layer are just the input features to the next layer. Therefore, on the one side, when pooling the input features to have spatial dimension of $1\times 1$, each reduced input feature channel naturally encodes holistic spatial information, and **to some extent may also implicitly encode some local information as the globally averaged result tends to be dominated by salient pixels**. On the other side, the length (i.e., $c_{in}$, the number of input feature channels) of the reduced input features is typically $>>$ the kernel size $k\times k$. Taking ResNet18 as an example, $c_{in}$ at its convolutional layers excluding the first layer is {64/64/64/64, 128/128/128/128, 256/256/256/256, 512/512/512/512}, while $k\times k$ is only 9. Therefore, $c_{in}$ reduced input feature channels should contain rich holistic spatial information, and to some extent may also implicitly encode more local information (compared to that for one single channel) due to the reason clarified above, compensating for the loss of local spatial information by global average pooling. This may be the underlying primary reason why a spatially pooled feature can still generate a spatial kernel attention well. **(2)** In the Appendix of the manuscript, we provided a set of experiments to compare the performance of adding ODConv to different layers of MobileNetV2 ($0.5\times$), showing adding ODConv to more layers brings more accuracy gain with slightly increased model complexity, please see Table 9. Because of this and following DyConv, we apply ODConv to all convolutional layers except the first layer, which should also compensate for local spatial information loss of the input features by global average pooling. **(3)** Based on deep CNNs, using input features of small size to predict larger-size output has become common in other computer vision tasks. For instance, in semantic image segmentation, a Fully Convolutional Network [3] takes a source image of arbitrary size as the input, sequentially uses a number of convolutional layers and several-staged (e.g., 5) down-sampling operations to extract discriminative features (the output features after the last down-sampling operation have a significantly reduced spatial size (e.g., $1/32$ of the source image size), and directly applies $1\times1$ convolutions (reshaped fully connected layers) to recover the feature size to be the same to that of the source image, and finally predicts the same-sized pixel-wise output. U-Net like structures as well as the encoder and decoder structures also incorporate somewhat similar designs. In single image super resolution, the problem is naturally defined to lift a low-resolution image to a corresponding high-resolution image with high quality, and deep CNNs are now the dominant solutions [4]. These examples show deep CNNs have great potentials to handle spatial information loss, which may be also useful for answering why in our ODConv a spatially pooled feature can still generate a spatial kernel attention well.
> > >
> > > We hope our above responses are helpful for a better understanding of our ODConv. Looking forward to hearing your feedback.
> > >
> > > Many thanks for your time and patience.
> > >
> > > [1] Matthew D Zeiler and Rob Fergus, "Visualizing and Understanding Convolutional Networks", ECCV  2014.
> > >
> > > [2] Robert Geirhos, Patricia Rubisch, Claudio Michaelis, Matthias Bethge, Felix A. Wichmann and Wieland Brendel, "ImageNet-trained CNNs are biased towards texture; increasing shape bias improves accuracy and robustness", ICLR 2019
> > >
> > > [3] Jonathan Long, Evan Shelhamer and Trevor Darrell, “Fully Convolutional Networks for Semantic Segmentation”, CVPR 2015.
> > >
> > > [4] Chao Dong, Chen Change Loy, Kaiming He and Xiaoou Tang “Learning a Deep Convolutional Network for Image Super-Resolution”, ECCV 2014.

---

> > > > ### Comment · Reviewer_PSFr · 2021-11-27
> > > > **response**
> > > >
> > > > Thanks for your response. I'm satisfied with your explanation.

---

> ### Author Response · Authors · 2021-11-22
> **Responses to Official Review by Reviewer PSFr: Part 1**
>
> Thank you so much for the detailed and constructive comments, and the recognition of the motivation, the novelty and the experiments of our work. Please see our below responses to your concerns one by one.
>
> 1.**To your first concern regarding a better understand of the proposed dynamic mechanism** “1. In Table 7, I observe the gain...the spatial dimension (70.25% vs 72.42%) is the largest...weaken the efficiency of the proposed model.” **and** “My first concern is critical...I hope the author to add some visualization or other analysis ... have a better understand of the proposed dynamic mechanism.”
>
> **Our responses** are: **(1)** Indeed, according to the first three results of Table 7, the spatial convolutional kernel attention $\alpha_{si}$ brings the largest gain to the baseline model compared to the other two attentions ($\alpha_{ci}$ and $\alpha_{fi}$ computed along the dimension of the input channels and the filters, respectively). Actually, we also observed similar performance trends with other backbone networks. Intuitively, at the first glance, it may easily lead to a weird feeling just as you mentioned the input features used to predict $\alpha_{si}$ (as well as the other three attentions) are always globally averaged, reducing their spatial dimension to one pixel. **However**, it should be noted that our ODConv uses the attentions generated from the input features to modify the convolutional kernels, which is quite different from popular self-calibration attention mechanisms (e.g., SE and CBAM use the channel attentions generated from the output features of a convolutional layer to recalibrate the output features themselves). As our ODConv is not a self-calibration attention module, reducing the spatial dimension of the input features to one pixel (but the length of the reduced vector is large enough) will not affect its promising performance. To validate this, we conduct another experiment on the ImageNet dataset with the ResNet18 backbone. In the experiment, for $\alpha_{si}$, we pool the input features to have a size of 3$\times$3 instead of 1$\times$1, and show the results in the below Table. It can be seen that preserving more spatial information only brings 0.06%|0.02% gain to top-1|top-5 accuracy, which well proves the effectiveness of our current design.
>
> | Models | Params | MAdds | Top-1 Acc (%) | Top-5 Acc (%) |
> | -- | :-------: | :--------: |  :--------: |  :--------: |
> | ResNet18 | 11.69M | 1.814G | 70.25 | 89.38 |
> | + ODConv ($1\times$, $r=1/4$, pooling to 1$\times$1) | 12.58M | 1.839G | 73.41 | 91.29 |
> | + ODConv ($1\times$, $r=1/4$, pooling to 3$\times$3)  | 14.85M | 1.842G | **73.47** | **91.31** |
>
> **(2)** As for $\alpha_{ci}$ (adding dynamic attentions to each input channel of every filter of a convolutional kernel), its effectiveness is due to the above described property of our ODConv, and is also due to its complementarity to the other three attentions $\alpha_{si}$, $\alpha_{fi}$ and $\alpha_{wi}$. Actually, from Table 7 it can be seen that $\alpha_{ci}$ performs better than $\alpha_{fi}$. Furthermore, applying our ODConv to more layers or successive layers of a network will help to improve the training performance, which can be clearly seen from the results of Table 9 in the Appendix of the original manuscript. **(3)** It should be noted that the default hyperparameter setting of our ODConv used in the main experiments (when compared with existing methods) is not the optimal one. There still leaves considerable room to further improve the performance for different backbone networks with our method, you are referred to **our responses to the first concern of Reviewer fTd9** for details. **(4)** Following your constructive suggestions, we use the well-trained ResNet18 models reported in Table 7 and all 50,000 images in the ImageNet validation set, and conduct a set of experiments to analyze the learnt attention values for $\alpha_{si}$, $\alpha_{ci}$, $\alpha_{fi}$, $\alpha_{wi}$ and their combination by providing lots of visualization examples and the statistical distributions across different layers. Detailed results are shown in Figure 5-6 (page 19-20 of the revised manuscript). We can get the following observations: **a)** Four types of attentions $\alpha_{si}$, $\alpha_{ci}$, $\alpha_{fi}$ and $\alpha_{wi}$ are complementary to each other by visualization examples, which echoes the conclusion from Table 7. The most salient attention will combat the failure cases of the other three attentions, helping the model to produce accurate predications. **b)** All four types of attentions demonstrate varying attention value distribution trends across different layers of the trained model, showing their capability to capture rich context cues. **c)** Each attention has its own value distribution trend, which is more diverse for $\alpha_{si}$ and $\alpha_{wi}$ than for $\alpha_{ci}$ and $\alpha_{fi}$, showing $\alpha_{si}$ and $\alpha_{wi}$ maybe more important to some degree.

---

### Official Review · Reviewer_kYmD · 2021-11-02

**Correctness:** 3
**Technical Novelty And Significance:** 2
**Empirical Novelty And Significance:** 3
**Recommendation:** 6
**Confidence:** 4

**Main Review:**

Pros)
+ The paper is written very well and easy to follow.
+ The idea is simple, and therefore the message is clear. The results look promising.
+ Experiments are well performed with popular competitors.

Cons)
- Involving orthogonal attentions seems complementary to each other, but any intuitions why decomposing like those work well compared with the naive ones (in dyconv and cordconv) are not clearly stated.
- Too many italics harm the readability


Comments)
1. Did the authors run all the experiments including the competitive methods in Table 1 and Table 2 by themselves?
2. Why ODConv is faster than DyConv in Table 8 (right)?
3. Please specify the training budget including speed compared with condconv and dyconv.
4. It would be better to report the model speed in Table 9 to confirm the speed improvements from diverse configurations.
5. Please specify the # parameters and FLOPs in Table 4. If possible, it would be nice to report the model accuracy as well.
6. In the COCO detection experiments, did the authors use ImageNet-pretrained models without dynamic convolutions or the pretrained models with dynamic convolutions reported in Table 1 and Table 2?
7. How does the model accuracy of CondConv used at every layer like the proposed method can be changed?
8. Can the authors visualize a_si, a_ci, a_fi, and a_wi to provide some insights? I am just curious about a trend of the learned attention vectors and which one is dominant.
9. The combination with the proposed method (additionally including dyconv and condconv) and SE-Net variants performing attention on a feature is likely to be complimentary. Can the authors show further improvement upon architectures incorporating a SE-like module?

**Summary Of The Paper:**

This work presents a dynamic convolution method based on Dynamic Conv (dyconv) and Coordinate Conv (cordconv) by incorporating more attention weights to the multiple convolutional kernels to mix kernels more effectively resulting in better accuracies on the ImageNet classification and COCO object detection tasks.

**Summary Of The Review:**

This paper is well written, and the results about the accuracy improvements look promising. A major drawback is the actual inference speed when compared with the competitors, but it would not be a matter because one can constrain the model to the target speed in a particular circumstance. I recommend the authors study more the way of speedup the model in practice for practitioners.

---

> ### Author Response · Authors · 2021-11-22
> **Responses to Official Review by Reviewer kYmD: Part 3**
>
> 9.**To your ninth question** “9. The combination with the proposed method (additionally including dyconv and condconv) and SE-Net variants performing attention on a feature is likely to be complimentary. Can the authors show further improvement upon architectures incorporating a SE-like module?”
>
> **Our responses** are: **(1)** Thanks for your constructive comment. Accordingly, we performed two sets of experiments on the ImageNet dataset with the MobileNetV3-Small and ResNet18 backbones. Note that the MobileNetV3-Small backbone is actually constructed with the SE module. For the ResNet18 backbone, we use its SE variant reported in Table 2. In the first set of experiments, we train the MobileNetV3-Small backbone with CondConv, DyConv, DCD and our ODConv separately, adopting the same training settings used for the MobileNetV2 backbones (see page 13 for experimental details). In the second set of experiments, we use ResNet18+SE to mainly test the potential of our ODConv. Detailed results are summarized in the below two Tables. **(2)** It can be seen that our ODConv brings promising improvements to these two backbones incorporating the SE module. Even for MobileNetV3-Small, a much stronger light-weight backbone network, our ODConv (4$\times$) obviously performs better than all counterparts, and our ODConv (1$\times$) with the smallest model size also outperforms 3 of 4 counterparts.
>
> | Models | Params | MAdds | Top-1 Acc (%) | Top-5 Acc (%) |
> | -- | :-------: |  :-------: | :--------: |  :--------: |
> | MobileNetV3-Small | 2.94M | 61.1M | 67.14 | 87.36 |
> | + CondConv (8$\times$) | 14.39M | 74.8M | 68.06 | 87.62 |
> | + DyConv (4$\times$) | 4.77M | 64.7M| 69.13 | 88.31 |
> | + DCD | 3.57M | 65.7M | 68.18 | 87.71 |
> | + ODConv (1$\times$) | 3.29M | 63.7M | 68.45 | 87.96 |
> | + ODConv (4$\times$) | 4.65M | 66.9M | **69.68** | **88.64** |
>
>
> | Models | Params | MAdds | Top-1 Acc (%) | Top-5 Acc (%) |
> | -- | :-------: | :--------: | :---------:  | :-----------: |
> | ResNet18 | 11.69M | 1.814G | 70.25 | 89.38 |
> | + SE | 11.78M | 1.816G | 70.98 | 90.03 |
> | + SE & ODConv (4$\times$) | 44.99M | 1.918G | **74.04** | **91.38** |
>
> 10.**To the first one of your mentioned two weaknesses** “Involving orthogonal attentions seems complementary to each other, but any intuitions why decomposing like those work well compared with the naive ones (in dyconv and cordconv) are not clearly stated.”
>
> **Our responses** are: **(1)** In the original manuscript, we discussed our basic intuitions when analyzing the main limitations of CondConv and DyConv (italic sentences of the Introduction section (page 2), and the paragraph titled **“Limitation Discussions”** of the Method section (page 4)), and when clarifying the design insights of our ODConv (the paragraph titled **“A Deep Understanding of ODConv”** of the Method section (page 4)). **(2)** Generally, our ODConv leverages a multi-dimensional attention mechanism with a parallel strategy to learn four types of attentions $\alpha_{si}$, $\alpha_{ci}$, $\alpha_{fi}$, $\alpha_{wi}$ for convolutional kernels along all dimensions of the kernel space at any convolutional layer. In principle, progressively multiplying these four types of attentions to the convolutional kernel in the location-wise, channel-wise, filter-wise and kernel-wise orders makes convolution operations be different w.r.t. all spatial locations, all input channels, all filters and all kernels for each input sample, providing a performance guarantee to capture rich context cues. **(3)** Following your constructive suggestion, we also provide a lot of visualization examples as well as the statistical distributions of learnt attention values for $\alpha_{si}$, $\alpha_{ci}$, $\alpha_{fi}$, $\alpha_{wi}$ and their combination across different layers of the trained model, showing more insights by experimental observations. Please see our responses to your eighth question.
>
> 11.**To the second one of your mentioned two weaknesses** “Too many italics harm the readability.”
>
> **Our responses** are: **(1)** In the original manuscript, italic descriptions (on page {2,4,5,6}) are used to highlight the motivation, the insights and the implementation of our ODConv, and the experimental settings for fair performance comparisons. **(2)** We are really sorry if these italics impact the readability. Accordingly, we remove most of them in the revised manuscript.
>
> 12.**To your comments regarding the drawback of our method** “A major drawback is the actual inference speed ... I recommend...the way of speedup the model in practice for practitioners.”
>
> **Our responses**: you are referred to **our responses to the second concern of Reviewer fTd9** for more discussions.
>
> **Finally**, regarding more experiments and discussions that we have made during the rebuttal phase, you are referred to our top-level comments titled **“The Summary of Our Responses to All Official Reviews”**, our responses to the other reviewers, and the revised manuscript.

---

> > ### Comment · Reviewer_kYmD · 2021-11-29
> > **Response to the rebuttal**
> >
> > Thank you for your great efforts to provide detailed responses. I am satisfied with most of the responses. Please revise the final manuscript by reflecting all the reviewers' comments.
> >
> > It would be good if the authors could further improve by optimizing the method to outperform the competitors reaching the baseline model in terms of speed. I would like to keep my score as weak accept.

---

> ### Author Response · Authors · 2021-11-22
> **Responses to Official Review by Reviewer kYmD: Part 2**
>
> 6.**To your sixth question** “6. In the COCO detection experiments, did the authors use ImageNet-pretrained models without dynamic convolutions or the pretrained models with dynamic convolutions reported in Table 1 and Table 2?
>
> **Our responses** are: **(1)** **We used both**. For each of two object detectors (Faster R-CNN and Mask R-CNN), the ImageNet-pretrained models without dynamic convolutions are used as the baselines, and the ImageNet-pretrained models with different dynamic convolution designs are used for performance comparison. **(2)** For fair comparisons, we followed the standard training and test settings in the downstream MS-COCO object detection experiments. Specifically, for each of two object detectors, we used the ResNet50|MobileNetV2 (1.0$\times$) model (with or without dynamic convolutions) pretrained on the ImageNet as the backbone, and used regular static convolutions in the other layers of the neck and the head. These were clearly stated in the paragraph titled **“Experimental Setup”** of the original manuscript (page 7).
>
> 7.**To your seventh question** “7. How does the model accuracy of CondConv used at every layer like the proposed method can be changed?”
>
> **Our responses** are: **(1)** Following your constructive suggestion, we conducted an experiment on the ImageNet dataset with the ResNet18 backbone. Following the same settings for DyConv and our ODConv, we apply CondConv to all convolutional layers except the first layer, and train the model. Detailed results are shown in the below Table. **(2)** It can be seen that this brings 0.38% improvement to top-1 accuracy (our ODConv (1$\times$) achieves 73.1% top-1 accuracy), which is consistent with the ablative results reported in the paper of CondConv.
>
> | Models | Params | MAdds | Top-1 Acc (%) | Top-5 Acc (%) |
> | -- | :-------: | :--------: | :---------:  | :-----------: |
> | ResNet18 | 11.69M | 1.814G | 70.25 | 89.38 |
> | + CondConv (8$\times$, to all conv layers except the first layer) | 92.19M | 1.909G | **72.37** | **90.49** |
> | + CondConv (8$\times$, to last 6 residual blocks (default setting)) | 81.35M | 1.894G | 71.99 | 90.27 |
>
>
> 8.**To your eighth question** “8. Can the authors visualize a_si, a_ci, a_fi, and a_wi to provide some insights? I am just curious about a trend of the learned attention vectors and which one is dominant.”
>
> **Our responses** are: **(1)** Thanks for your constructive suggestion. Accordingly, we conducted a set of experiments to analyze the learnt attention values for $\alpha_{si}$, $\alpha_{ci}$, $\alpha_{fi}$, $\alpha_{wi}$ and their combination by providing lots of visualization examples and the statistical distributions across different layers. In the experiments, we apply the well-trained ResNet18 models with ODConv using different attention combinations (reported in Table 7) to all 50,000 images in the ImageNet validation set. Detailed results are shown in Figure 5-6 (page 19-20 of the revised manuscript). We can get the following observations: **a)** Four types of attentions $\alpha_{si}$, $\alpha_{ci}$, $\alpha_{fi}$ and $\alpha_{wi}$ are complementary to each other by visualization examples, which echoes the conclusion from Table 7. The most salient attention will combat the failure cases of the other three attentions, helping the model to produce accurate predications. **b)** All four types of attentions demonstrate varying attention value distribution trends across different layers of the trained model, showing their capability to capture rich context cues. **c)** Each attention has its own value distribution trend, which is more diverse for $\alpha_{si}$ and $\alpha_{wi}$ than for $\alpha_{ci}$ and $\alpha_{fi}$, showing $\alpha_{si}$ and $\alpha_{wi}$ maybe more important to some degree. **(3)** Note that the dimension of attentions $\alpha_{si}$, $\alpha_{ci}$, $\alpha_{fi}$ and $\alpha_{wi}$ is quite different, e.g., 9 for $\alpha_{si}$ with $3\times3$ convolutional kernels, 4 for $\alpha_{wi}$ with $n=4$, and typically several hundred for $\alpha_{ci}$ and $\alpha_{fi}$. This is another perspective why they are different and complementary in their design nature regarding all four dimensions of the convolutional kernel space for any convolutional layer.

---

> ### Author Response · Authors · 2021-11-22
> **Responses to Official Review by Reviewer kYmD: Part 1**
>
> Thank you so much for the detailed and constructive comments, and the recognition of the writing, the idea and its performance, and the experiments of our work. Please see our below responses to your questions and concerns one by one.
>
> 1.**To your first question** “1. Did the authors run all the experiments including the competitive methods in Table 1 and Table 2 by themselves?”
>
> **Our responses** are: **(1)** Yes, all the experiments including the reference methods in Table 1, Table 2 and the other Tables were run by us using the public codes and the popular training and test settings used in the community. For clean and fair performance comparisons, we also did not use advanced training tricks such as Mixup and Label Smoothing. These were clearly stated in the paragraph titled **“Experimental Setup”** of the original manuscript (page 6). **(2)** We will release the whole source code package of our work in the near future, and hope it could advance the research in dynamic convolution.
>
> 2.**To your second question** “2. Why ODConv is faster than DyConv in Table 8 (right)?”
>
> **Our responses** are: **(1)** We think you refer to our ODConv (1$\times$). ODConv (1$\times$) is faster than DyConv because the number of convolutional kernels $n$ is 4 for DyConv while $n=1$ for ODConv (1$\times$). We revised the manuscript to make it clearer. **(2)** When $n=4$, it can be seen that our ODConv (4$\times$) is slower than DyConv (4$\times$).
>
> 3.**To your third question** “3. Please specify the training budget including speed compared with condconv and dyconv.”
>
> **Our responses** are: **(1)** Following your constructive suggestion, we conducted a set of experiments to compare the training budget of our ODConv, CondConv, DyConv and DCD. Specifically, experiments are performed on the ImageNet dataset with the ResNet50 and MobileNetV2 backbones, using the same training settings as in the manuscript. Regarding the training budget, we report results in terms of three metrics (seconds per batch, minutes per epoch, and the total number of hours for the whole training) in the below two Tables. **(2)** Regarding different dynamic convolution methods, it can be seen that the training budget trend is somewhat similar to that for the runtime inference speed (Table 8 of the manuscript). **(3)** Generally, the training budget for dynamic convolution methods is obviously heavier (about 2-3$\times$) than that for the baseline model, which is mainly due to the time-intensive back-propagation process for additive convolutional kernels weighted with input-dependent attentions.
>
> | Models | Batch Cost (second) | Epoch Cost (minute) | Total Cost (hour) |
> | -- | :-------: | :--------: | :--------: |
> | ResNet50 | 0.216 | 18.4 | 31.7 |
> | + CondConv (8$\times$) | 0.429 | 36.4 | 62.8 |
> | + DyConv (4$\times$) | 0.569 | 48.0 | 82.3 |
> | + DCD | 0.414 | 35.1 | 60.4 |
> | + ODConv (1$\times$) | 0.513 | 43.4 | 73.5 |
> | + ODConv (4$\times$) | 0.658 | 54.9 | 92.6 |
>
>
> | Models | Batch Cost (second) | Epoch Cost (minute) | Total Cost (hour) |
> | -- | :-------: | :--------: | :--------: |
> | MobileNetV2 (1.0$\times$) | 0.113 | 9.7 | 25.3 |
> | + CondConv (8$\times$) | 0.175 | 15.1 | 39.3 |
> | + DyConv (4$\times$) | 0.310 | 26.4 | 67.9 |
> | + DCD | 0.282 | 24.0 | 61.9 |
> | + ODConv (1$\times$) | 0.301 | 25.6 | 65.7 |
> | + ODConv (4$\times$) | 0.402 | 34.0 | 86.8 |
>
> 4.**To your fourth question** “4. It would be better to report the model speed in Table 9 to confirm the speed improvements from diverse configurations.”
>
> **Our responses** are: **(1)** Following your constructive suggestion, we tested the runtime speed for each configuration, and added the results to Table 9 of the revised manuscript. **(2)** It can be seen that adding ODConv to more layers introduces more extra latency to the runtime inference speed.
>
> 5.**To your fifth question** “5. Please specify the # parameters and FLOPs in Table 4. If possible, it would be nice to report the model accuracy as well.”
>
> **Our responses** are: **(1)** In Table 4, we compare the generalization ability of different dynamic convolution methods in the downstream MS-COCO object detection task, following the standard training and test settings. Note that the ResNet50|MobileNetV2 (1.0$\times$) model (based on each of four dynamic convolution designs) pretrained on the ImageNet is used as the backbone for each object detector, and the other layers of the neck and head still use regular convolutions. Therefore, the accuracy (top-1 and top-5), the number of parameters, and the FLOPs of the pretrained ResNet50|MobileNetV2 model for different dynamic convolution methods are the same to those reported in Table 1 and Table 2. For the neck and head of each object detector, the number of parameters and the FLOPs are fixed as different dynamic convolution methods are only used in the pretrained backbone models. **(2)** Following you suggestion, we added the corresponding results to Table 4 of the revised manuscript.

---

### Official Review · Reviewer_CKv4 · 2021-11-02

**Correctness:** 3
**Technical Novelty And Significance:** 4
**Empirical Novelty And Significance:** 3
**Recommendation:** 8
**Confidence:** 3

**Main Review:**

Highlights:
1. The authors provide a detailed analysis of dynamic convolution operations and reveal the limitation of previous works.
2. Based on the analysis, the authors propose omni-dimensional dynamic convolution (ODConv), where multiple attention layers are employed to generate attention weights of different convolution dimensions.
3. The proposed method can be plugged into most existing CNN architectures and has good performance on various public datasets, such as ImageNet and MS-COCO.
4. The technical part is easy to follow, and the experimental part is comprehensive.

Drawbacks:
Actually not too much. The following are just some minor questions:
1. As shown in Table {1,2,3}, the improvement brought by ODConv decreases as the model size increases. So it seems that the improvement of strong baselines (#param > 100M) would be limited.
2. Although advanced training tricks are not necessary for clean performance comparisons, it would be better to have a study about whether ODConv can also work well when using heavy data augmentations and a longer training schedule.


**Summary Of The Paper:**

This work proposes a dynamic convolution that is equipped with attention layers in all dimensions. Extensive experiments show that the proposed dynamic convolution can yield a more powerful representation than counterparts, especially in light-weight backbones.

**Summary Of The Review:**

In summary, this work presents a new dynamic convolution that can provide a powerful representation for image classification and object detection. The paper is well written and properly structured, and the extensive experiments prove the effectiveness of the proposed method.

---

> ### Author Response · Authors · 2021-11-22
> **Responses to Official Review by Reviewer CKv4**
>
> Thank you so much for the detailed and constructive comments, and the recognition of our work including the motivation, the proposed method, the writing and the experiments. Please see our below responses to your questions one by one.
>
> 1.**To your first question** “As shown in Table {1,2,3}, the improvement brought by ODConv decreases as the model size increases. So it seems that the improvement of strong baselines (#param > 100M) would be limited.”
>
> **Our responses**: **(1)** Yes, for different backbone networks (baselines), the performance improvement brought by our ODConv is not at the same level. On the ImageNet dataset (as shown in Table {1,2,3}), the performance improvement of ODConv indeed tends to decrease as the backbone network becomes stronger. Actually, the similar performance improvement trend can be also observed in the experiments on the MS-COCO object detection dataset, please see Table 4. **(2)** This is something common for existing methods based on dynamic convolutions or others. From Table {1,2,3} you can also find that DyConv, CondConv and DCD demonstrate the similar performance improvement trend as our method. This is mainly due to the increased network depth and the increased number of parameters, making strong backbone networks already have much higher performance compared to the ones with the smaller model size. **(3)** Although the performance improvement of our ODConv obviously decreases as the baseline model becomes very deep and large, **it is still decent but not limited**. From Table 3 it can be seen that our ODConv (1$\times$) brings **1.57%** top-1 gain to the ResNet101 backbone on the ImageNet dataset, and the top-1 gain by ODConv (2$\times$) reaches up to **1.86%**, outperforming reference methods with clear margins (note that CondConv (8$\times$) and DyConv (4$\times$) failed in training the ResNet101 model due to the huge memory cost (regarding the computational resources we have), and even on the ResNet50 backbone, CondConv (8$\times$), DyConv (4$\times$) and DCD show less than 1% top-1 gain as can be seen from Table 2).
>
> 2.**To your second question** “Although advanced training tricks are not necessary for clean performance comparisons, it would be better to have a study about whether ODConv can also work well when using heavy data augmentations and a longer training schedule.”
>
> **Our responses** are: **(1)** Following your constructive suggestion, we perform a set of experiments on the ImageNet dataset with the ResNet18 backbone trained with ODConv (4$\times$). In the experiments, we set r=1/16 for our method, first train ResNet18 with Label Smoothing, Mixup and {Label Smoothing + Mixup} separately for 100 epochs (the other training settings are the same to those used for Table 2 of the manuscript), and then train ResNet18 with {Label Smoothing + Mixup} for 120 epochs instead of 100 epochs. Detailed results are summarized in the below Table. It can be seen that our ODConv works well when using aggressive data augmentations and a longer training schedule, showing further improved performance. **(2)** In the Appendix of the original manuscript, we actually provided an experimental comparison on the ImageNet dataset with the MobileNetV2 (1.0$\times$, 0.75$\times$, 0.5$\times$) backbones trained with the increased training schedule (from 150 to 300 epochs), showing consistently improved performance for both ODConv (1$\times$) and ODConv (4$\times$), please see Table 13 vs. Table 1.
>
> | Models | Top-1 Acc (%) | Top-5 Acc (%) |
> | -- | :-------: | :--------: |
> | ResNet18 + ODConv (4$\times$) | 73.97 | 91.35 |
> | + Label Smoothing | 74.15 | 91.63 |
> | + Mixup | 74.05 | 91.42 |
> | + Label Smoothing & Mixup | 74.18 | 91.68 |
> | + Label Smoothing & Mixup & Longer Training Schedule | **74.59** | **91.74** |
>
> **Finally**, regarding more experiments and discussions that we have made during the rebuttal phase, you are referred to our top-level comments titled **“The Summary of Our Responses to All Official Reviews”**, our responses to the other reviewers, and the revised manuscript.

---

> > ### Comment · Reviewer_CKv4 · 2021-11-28
> > **Comments on reply of the authors**
> >
> > Thanks for providing the reply based on our reviews.
> >
> > I look through all replies and they've addressed my concerns. I think this omni-dimensional dynamic convolution is to the interests of many researchers in the computer vision community. I choose to keep my rating and recommend an acceptance to this paper.

---

### Official Review · Reviewer_fTd9 · 2021-11-03

**Correctness:** 4
**Technical Novelty And Significance:** 3
**Empirical Novelty And Significance:** Not applicable
**Recommendation:** 8
**Confidence:** 4

**Main Review:**

First, I think the topic of the work is an interesting and valuable one. The two ideas that this work borrows from prior methods, (1) filter recalibration with attention in SENet and (2) additive kernels in CondConv/DyConv, are both well-received and are known to bring meaningful improvements to base architectures. Consequently, an effort to combine and generalize these ideas is indeed intriguing and may bring further improvements and insights.

Regarding the technical details of the approach, it's heavily based on prior work, particularly the design of the attention computation. The changes and additions are justified well. The method is overall quite efficient without too much overhead, due to the separate attention weights for different dimensions.

The experiments are quite comprehensive, covering multiple backbones, reference methods, and tasks. It is particularly welcome that they help provide a complete picture of the tradeoffs by also comparing the model sizes and MAdds. Overall, the results indicate that the lighter model (ODConv 1x) offers comparable or slightly better accuracies with less compute, while the larger model (ODConv 4x) provides consistent accuracy improvements. The authors also provide a wide range of ablation studies, which help a better understanding of various aspects necessary when deploying the technique to other applications.

A slight shortcoming is that it appears there might be better combinations of the hyperparameters, at least for certain scenarios, that are missed. This is understandably due to the many factors that are involved. For example, it was concluded that "r=1/16 strikes the best tradeoff between model accuracy and efficiency" (page 8). However, increasing r to 1/4 almost closes the gap to ODConv 4x while adding a much smaller overhead. Attention sharing, investigated in Tab. 12, is another such opportunity for better accuracy with little overhead compared to using more kernels.

The author does not discuss any potential limitations or disadvantages compared to reference methods. Is it the conclusion that ODConv offers improvements over competing methods, e.g., CondConv, DyConv, SENet, without any drawbacks? Some insights in this regard can help guide applications.

Another discussion or analysis I think can be valuable is the comparison with dynamic-filter/kernel prediction networks. Since all dimensions are now being recalibrated, the approach is closer to those types of methods. For example, ODConv can be seen as a separable version of predicting full kernels.

**Summary Of The Paper:**

The authors present ODConv, a type of dynamic convolutional operation. ODConv combines two prior ideas, i.e. (1) filter recalibration with attention in SENet and (2) additive kernels in CondConv/DyConv, and also generalizes to all remaining dimensions of convolutional filters. The authors propose to use ODConv as drop-in replacements for regular convolutions in standard CNNs. Experiments and analysis over several tasks demonstrate that ODConv has noticeable advantages over alternatives and offers a good tradeoff between performance and compute.

**Summary Of The Review:**

The idea and technical approach are well motivated and justified. It connects and generalizes two important prior ideas and offers an elegant solution. The results are convincing as the experiments show consistent advantages over a range of tasks and base architectures. The work also has good potential in practical applications due to its convenience as a drop-in replacement for standard convolutions.

---

> ### Author Response · Authors · 2021-11-22
> **Responses to Official Review by Reviewer fTd9: Part 2**
>
> 3.**To your comment** “Another discussion or analysis I think can be valuable is the comparison with dynamic-filter/kernel prediction networks. Since all dimensions are now being recalibrated, the approach is closer to those types of methods. For example, ODConv can be seen as a separable version of predicting full kernels.”
>
> **Our responses** are: **(1)** We fully agree that our work is also related to the works using dynamic weights such as Dynamic Filter Networks, Kernel Predication Networks, Hypernetworks, MetaNet, WeightNet, CGC, WE and so forth, as they all share the similarity of using the dynamic weight mechanism to improve the capacity of a baseline network. Actually, we had a discussion of this lined research and our work in the original manuscript, please see the third paragraph (titled “**Dynamic Weight Networks**”, page 3) of the Related Work section. Furthermore, in Table 2, we also provided the experimental comparison of our ODConv with WeightNet, CGC and WE on the ImageNet dataset with the ResNet18 and ResNet50 backbones, showing that our ODConv (1$\times$)|ODConv (4$\times$) outperforms them by obvious margins (over 1.49%|2.36% top-1 margin for the ResNet18 backbone, and over 0.44%|1.00% top-1 margin for the ResNet50 backbone). As Dynamic Filter Networks and Kernel Predication Networks are primarily used for image processing tasks, we did not consider them in our experiments focused on image classification and object detection tasks. **(2)** Generally, regarding the aforementioned works for “Dynamic Weight Networks”, the weights of a baseline network are either generated from another network or modified by the attentive modules. That is, the number of convolutional parameters is fixed to the baseline network. In sharp contrast to them, our ODConv smartly connects and generalizes the attentive weight recalibration and the additive convolutional kernels with a neat omni-dimensional attention module (which learns complementary attentions for all four dimensions of the convolutional kernel space parallelly), bringing a more powerful drop-in dynamic convolution design. This is the key difference of our ODConv against them.
>
> **Finally**, regarding more experiments and discussions that we have made during the rebuttal phase, you are referred to our top-level comments titled **“The Summary of Our Responses to All Official Reviews”**, our responses to the other reviewers, and the revised manuscript.

---

> ### Author Response · Authors · 2021-11-22
> **Responses to Official Review by Reviewer fTd9: Part 1**
>
> Thank you so much for the thorough and constructive comments, and the recognition of our work including the topic, the proposed method, the presentation, the experiments and the potential applications. Please see our below responses to your concerns (about a slight shortcoming and more discussions in two aspects) one by one.
>
> 1.**To your comment regarding there may exist better combinations of the hyperparameters** “A slight shortcoming is that it appears there might be better combinations of the hyperparameters, at least for … This is understandably due to the many factors that are involved. For example… "r=1/16 strikes the best tradeoff between model accuracy and efficiency" (page 8). However, increasing r to 1/4 almost closes the gap to ODConv 4x while adding a much smaller overhead. Attention sharing, investigated in Tab. 12, is another such opportunity for better accuracy with little overhead compared to using more kernels.”
>
> **Our responses** are: **(1)** Yes, regarding the model accuracy, the default combination of the hyperparameters for our ODConv is not the optimal setting, even for the ResNet18 backbone used in your mentioned ablative experiments. We apply this setting to all backbone networks mainly for a better accuracy-efficiency (in terms of the model size) tradeoff and easy implementation, already showing obviously superior performance over reference methods. **(2)** For our ODConv, there actually exist a large number of combinations of the hyperparameters (including layer locations for placing ODConv, four types of convolutional kernel attentions $\alpha_{si}$, $\alpha_{ci}$, $\alpha_{fi}$ and $\alpha_{wi}$, the number of convolutional kernels $n$, the reduction ratio $r$ and the activation functions). Although we provided a wide range of ablation studies to show how they affect the performance of our ODConv, there still leaves considerable room to further improve the performance for different backbone networks if tuning the combination of these hyperparameters for each network. Specifically, besides your mentioned two experiments (increasing $r$ to 1/4 and w/o attention sharing), we better showed this potential in Table 7 of the original manuscript when studying the complementarity of four types of convolutional kernel attentions $\alpha_{si}$, $\alpha_{ci}$, $\alpha_{fi}$ and $\alpha_{wi}$. In the experiments, we set $r=1/4$ (as stated in Table 7), and the ResNet-18 model trained with ODConv (4$\times$) shows 74.33% top-1 accuracy, outperforming the model trained with our default setting by **0.36%** margin. In our recent experiments, when combing this setting with the non-shared attention strategy studied in Table 12, the top-1 accuracy of ResNet18 can be even boosted to **74.55%**. **(3)** We will release the whole source code package of our work in the near future, and hope the community would make a more deep and wide investigation of its potentials.
>
> 2.**To your comment regarding the discussion of potential limitations of our method** “The author does not discuss any potential limitations or disadvantages compared to reference methods. Is it the conclusion that ODConv offers improvements over competing methods, e.g., CondConv, DyConv, SENet, without any drawbacks? Some insights in this regard can help guide applications.”
>
> **Our responses** are: we follow your constructive suggestion and analyze the limitations of our work in three aspects. **(1)** According to the experimental results shown in Table{1, 2, 3, 8, 13-15, 18}, although our ODConv shows obviously better performance under the similar model size (e.g., ODConv (4$\times$) vs. DyConv (4$\times$)), it leads to slightly increased FLOPs and introduces small extra latency to the runtime inference speed, as the proposed four types of convolutional kernel attentions $\alpha_{si}$, $\alpha_{ci}$, $\alpha_{fi}$ and $\alpha_{wi}$ introduce a bit more learnable parameters. Under the similar model size, the training cost of our ODConv is also somewhat heavier than reference methods (e.g., ODConv (4$\times$) vs. DyConv (4$\times$)), please see newly added experimental results shown in Table 18 (page 18) of the revised manuscript. **(2)** As we discussed in the responses to your comment regarding there may exist better combinations of the hyperparameters, although we provided a wide range of ablative studies to analyze the effect of different hyperparameters of our ODConv using the ResNet18 backbone with the ImageNet dataset, and applied the resulting combination of the hyperparameters to all backbone networks, it is not the optimal setting to different backbone networks. This suggests the way to reduce the cost of our method, finding a reasonable combination to constrain the final model to the target speed and accuracy for a particular circumstance. **(3)** Besides, the potential of applying ODConv to more deep and large backbones beyond ResNet101 has not been explored due to the constraint of our available computational resource.

---

### Author Response · Authors · 2021-11-22
**The Summary of Our Responses to All Official Reviews**

Dear Reviewers, Area Chairs and Program Chairs,

We sincerely thank all four reviewers for their thorough and constructive comments. We are so excited that the motivation, the novelty, the writing and the experiments of our work have been well recognized by all four reviewers.

In the past about ten days, we carefully improved the experiments (using all computational resources we have), the clarifications and the discussions of our work to address the concerns, the questions and the requests by all four reviewers. **Summarily, we made the following improvements**:

**(1)** To further improve the experiments, we follow the constructive suggestions from each reviewer, and provide more experiments including **a)** a set of experiments to show that our ODConv can also bring promising accuracy improvements to the backbones (MobileNetV3-Small and ResNet18) incorporating the SE module; **b)** a set of experiments to show that the performance of our ODConv can be further improved when using heavy data augmentations (label smoothing and mixup) and a longer training schedule; **c)** a set of experiments to compare the training cost for our method and its counterparts (CondConv, DyConv and DCD); **d)** some other results and clarifications for better ablation studies.

**(2)** To have a better understanding of the proposed attention mechanism regarding four types of convolutional kernel attentions $\alpha_{si}$, $\alpha_{ci}$, $\alpha_{fi}$ and $\alpha_{wi}$ in our ODConv, we follow the constructive suggestions by Reviewer kYmD and Reviewer PSFr, and provide a lot of visualization examples as well as the statistical distributions of learnt attention values for $\alpha_{si}$, $\alpha_{ci}$, $\alpha_{fi}$, $\alpha_{wi}$ and their combination across different layers of the trained model, showing more insights by experimental observations.

**(3)** Regarding the performance trend of our method on the backbone networks with different capacities (concerned by Reviewer CKy4), we provide related discussions.

**(4)** Regarding the limitations of our method (requested by Reveiwer fTd9), we provide related discussions.

**(5)** We also provide detailed responses to the other concerns/questions/requests raised by each reviewer one by one.

**Finally**, based on the constructive comments by all four reviewers and our responses, **we carefully revised and updated the manuscript of our work**. We will release the whole source code package of our work in the near future, and hope it could advance the research in dynamic convolution. We hope our detailed responses and the updated manuscript are helpful to address the concerns, the questions and the requests of all reviewers.

---

### Public Comment · ~Zejiang_Hou1 · 2022-03-09
**Missing citation**

The idea, using multi-dimensional attention along the spatial, input channel, and output channel dimensions for dynamic convolution, was already proposed by "Parameter Efficient Dynamic Convolution via Tensor Decomposition" (PEDConv), published last year in BMVC 2021.

Equation (2) in your current version resembles Equation (1) in "Parameter Efficient Dynamic Convolution via Tensor Decomposition".
And the conclusions in your paper, e.g., "one single kernel can compete with or outperform existing dynamic convolution counterparts with multiple kernels" were also demonstrated by "Parameter Efficient Dynamic Convolution via Tensor Decomposition".

Given the similarities of ideas and formulations between the two papers, it would be better if the authors cite this closely related work and have a discussion on what are the new contributions of ODConv compared to PEDConv.

Thank you,

---

> ### Public Comment · ~Anbang_Yao1 · 2022-03-10
> **Re: Missing citation**
>
> Many thanks for sharing the work of PEDConv, Zejiang.
>
> PEDConv and our ODConv are obviously contemporaneous researches. Note that the submission deadline of ICLR 2022 (see https://iclr.cc/Conferences/2022/Dates) is even weeks before the paper decision notification date of BMVC 2021 (see https://www.bmvc2021-virtualconference.com/dates/), and it seems that PEDConv is not available on arXiv. The official ICLR 2022 guidelines (see https://iclr.cc/Conferences/2022/ReviewerGuide / https://iclr.cc/Conferences/2022/AuthorGuide ) clearly state "We consider papers contemporaneous if they are published (available in online proceedings) within the last four months. That means, since our full paper deadline is October 5, if a paper was published (i.e., at a peer-reviewed venue) on or after June 5, 2021, authors are not required to compare their own work to that paper."
>
> Furthermore, PEDConv and ODConv are different in formulation and design, although they happen to advance dynamic convolution research via improving the attention scheme. Similar to Dynamic Convolution Decomposition (DCD, published in ICLR 2021, which was discussed and thoroughly compared in our submission), PEDConv also uses a tensor decomposition strategy for the parameterization of a single shared kernel instead of multiple additive kernels. In contrast, our ODConv is a generalized dynamic convolution design, which leverages a multi-dimensional attention module with a parallel strategy to directly learn complementary attentions for additive convolutional kernels along all dimensions of the kernel space at any convolutional layer, without need of tensor decomposition and enjoying neat implementation. Our single kernel design is just a special case of ODConv.
>
> Thank you.

---

### Decision · Program_Chairs · 2022-01-20

**Decision:**

Accept (Spotlight)

**Comment:**

This paper presents ODConv, a convolution pattern which uses attention in the convolutions across all dimensions of the weight tensor.
The paper is well motivated and well explained, easy to follow.
This work is built on top of previous work, but reviewers all agree that the contributions of this paper are significant.
The experimental section is comprehensive, with several benchmarks, and show clear improvements.
The reviewers suggested a few additional remarks, and discussions to add to the paper, which the authors have addressed in the rebuttal. Reviewers seem in general happy with the authors answers to their concerns.
This seems like a sound and meaningful paper. I am fully in favour of acceptance, and I recommend this paper to be presented as a spotlight.